# Identification of the *Pseudomonas aeruginosa* AgtR-CspC-RsaL pathway that controls Las quorum sensing in response to metabolic perturbation and *Staphylococcus aureus*

Junze Qu[1], Liwen Yin[1], Shanhua Qin[1], Xiaomeng Sun[1], Xuetao Gong[1], Shouyi Li[2], Xiaolei Pan[3], Yongxin Jin[1], Zhihui Cheng[1], Shouguang Jin[1], Weihui Wu[1]*

1 Department of Microbiology, State Key Laboratory of Medicinal Chemical Biology, Key Laboratory of Molecular Microbiology and Technology of the Ministry of Education, College of Life Sciences, Nankai University, Tianjin, China, 2 Pancreas Center, Tianjin Medical University Cancer Institute and Hospital, National Clinical Research Center for Cancer, National Key laboratory of Druggability Evaluation and Systematic Translational Medicine, Tianjin Key laboratory of Digestive Cancer, Tianjin's Clinical Research Center for Cancer, Tianjin, People's Republic of China, 3 Department of Immunology, School of Basic Medical Sciences, Tianjin Medical University, Tianjin, China

* wuweihui@nankai.edu.cn

## Abstract

Environmental metabolites and metabolic pathways significantly influence bacterial pathogenesis and interspecies competition. We previously discovered that a mutation in the triosephosphate isomerase gene, *tpiA*, in *Pseudomonas aeruginosa* led to defective type III secretion and increased susceptibility to aminoglycoside antibiotics. In this study, we found that the *tpiA* mutation enhances the Las quorum sensing system due to reduced translation of the negative regulator RsaL. Further investigations demonstrated an upregulation of CspC, a CspA family protein that represses *rsaL* translation. DNA pull-down assay, along with genetic studies, revealed the role of AgtR in regulating *cspC* transcription. AgtR is known to regulate pyocyanin production in response to N-acetylglucosamine (GlcNAc), contributing to competition against *Staphylococcus aureus*. We demonstrated that CspC activates the Las quorum sensing system and subsequent pyocyanin production in response to GlcNAc and *S. aureus*. Overall, our results elucidate the AgtR-CspC-RsaL-LasI pathway that regulates bacterial virulence factors and its role in competition against *S. aureus*.

## Author summary

*Pseudomonas aeruginosa* is a Gram-negative bacterium notorious for causing a wide range of infections, particularly in hospitals. It thrives in moist environments and can lead to pneumonia, urinary tract and wound infections. The bacterium harbours multiple virulence factors and is able to replicate in a variety of niches in response to environmental signals, which contribute to its pathogenicity. Understanding of the bacterial metabolic and regulatory networks will provide clues for the development of novel therapeutic

**Data availability statement:** All relevant data are within the manuscript and its Supporting Information files.

**Funding:** This research was supported by National Natural Science Foundation of China (32170177 to WW, 32170199 to ZC), and the Science and Technology Committee of Tianjin (22JCYBJC00790 to ZC, 22JCYBJC00560 to YJ), the Fundamental Research Funds for the Central Universities, Nankai University (2122021405 to WW, 63231048 to ZC, 63241535 to YJ), the Open Fund of Ministry of Education Key Laboratory of Molecular Microbiology and Technology, Naikai University (NKU-KLMMTME-KFKT-202302 to SL). The funders had no role in study design, data collection and analysis, decision to publish, or preparation of the manuscript.

**Competing interests:** The authors have declared that no competing interests exist.

medicines and strategies. We previously found that mutation in a *P. aeruginosa* metabolic enzyme gene *tpiA* reduces the expression of an important virulence determinate, the type III secretion system. Here in this study, we demonstrate that *tpiA* mutation enhances the quorum sensing (QS) systems that play important roles in bacterial virulence. We further elucidated the regulatory pathway of TpiA mediated regulation on the QS systems. In addition, we demonstrated a role of the pathway in the bacterial competition with *Staphylococcus aureus*, a pathogenic bacterium that often co-exists with *P. aeruginosa* in chronic polymicrobial infections. Our results provide new insights into the regulation of virulence factors in *P. aeruginosa*.

## Introduction

*Pseudomonas aeruginosa* is a prevalent opportunistic Gram-negative pathogen that causes various acute and chronic infections in humans, particularly in patients with burn wounds, cystic fibrosis (CF), and compromised immunity [1–3]. The bacterium possesses an array of virulence factors that facilitate colonization, dissemination, and persistence. Pyocyanin, a phenazine pigment synthesized by approximately 95% of *P. aeruginosa* strains, is responsible for the characteristic blue-green color of these bacteria [4]. Pyocyanin exerts toxic effects against other bacterial species, fungi, and mammalian cells by inhibiting the electron transport chain, leading to the generation of reactive oxygen species [5,6]. Thus, pyocyanin plays a critical role in bacterial virulence and interspecies competition [7,8].

*P. aeruginosa* harbors two nearly identical operons, *phzA1-G1* (*phz1*) and *phzA2-G2* (*phz2*), which encode proteins that catalyze the synthesis of phenazine-1-carboxylic acid from chorismic acid. Phenazine-1-carboxylic acid is further converted to pyocyanin by PhzM and PhzS [9]. The expression of pyocyanin synthesis genes is regulated by quorum sensing (QS) systems [10]. QS is a bacterial cel-cell communication system that enables cells to respond to changes in cell density. [11–13]. *P. aeruginosa* possesses three central QS systems: Las, Rhl, and PQS. LasI and RhlI are responsible for synthesizing the signal molecules *N*-3-(oxo-dodecanoyl-homoserine) lactone (3-oxo-$C_{12}$-HSL) and *N*-butyryl-homoserine lactone ($C_4$-HSL), which are recognized by the regulatory proteins LasR and RhlR, respectively [14,15]. The PQS system includes the regulatory protein PqsR and two signaling molecules, 2-heptyl-3-hydroxy-4-quinolone (PQS) and 2-heptyl-4-quinolone (HHQ), synthesized by proteins encoded by the *pqsH* gene, the *pqsABCDE*, and *phnAB* operons [16,17]. These three QS systems are hierarchically interconnected, with the Las system positively regulating the Rhl and PQS systems [18,19]. Meanwhile, LasR directly activates the expression of *rsaL*, which encodes a DNA-binding protein that inhibits *lasI* transcription by binding to its promoter region, thus forming a negative feedback loop [20].

In addition to the central regulators and cognate signals, QS systems are modulated by various regulatory pathways and environmental signals [21,22]. For instance, the transcription of *lasR* is directly regulated by the global regulator Vfr and a β-lactam responsive regulator, AmpR [23,24]. ChIP-seq analysis has revealed that the sigma factor RpoN binds to the promoter regions of *lasI*, *rhlI*, and *pqsR* [25]. Under phosphate-depletion conditions, a two-component regulatory system, PhoR-PhoB, activates the expression of *lasI*, *pqsA*, *rhlR*, and *mvfR* [26]. In response to the peptidoglycan component N-acetylglucosamine (GlcNAc), the regulator AgtR enhances pyocyanin production. This response of *P. aeruginosa* promotes its pathogenesis and competition against *Staphylococcus aureus* during polymicrobial infections [27,28]. AgtR and the sensor protein AgtS form a two-component system. Upon sensing

δ-aminovalerate, γ-aminobutyrate and β-alanine, AgtS-AgtR activates the *agtABCD* operon encoding an ABC transporter system that is required for the uptake of the substrates [29]. However, the AgtS-AgtR mediated regulatory pathway on pyocyanin production remains to be elucidated.

The QS systems are also influenced by bacterial metabolism [21,22]. The catabolite repression regulator Crc has been found to regulate multiple QS genes [30,31]. A subsequent study revealed that Crc suppresses the translation of the Lon protease, which controls the stability of RhlI [32]. Additionally, PrsA represses the transcription of *lasR* in response to oleic acid [33]. We previously demonstrated that PvrA activates PQS signal synthesis genes in the presence of palmitic acid [34]. When *P. aeruginosa* was grown in whole blood from trauma patients, malonate utilization genes were upregulated compared to cells grown in whole blood from healthy individuals [35]. Further studies indicated that, compared to glycerol, malonate represses the expression of *lasI*, *lasR*, *rhlI*, and *rhlR* [36]. During chronic lung infections caused by *P. aeruginosa*, macrophages and monocytes release itaconate into the airways [37,38]. We have demonstrated that itaconate induces the deacetylation of the RNA-binding protein CspC, leading to the repression of *rsaL* translation, and subsequent upregulation of the Las system [39].

In *P. aeruginosa*, triosephosphate isomerase (TpiA) catalyzes the reversible conversion between glyceraldehyde-3-phosphate (G3P) and dihydroxyacetone phosphate (DHAP). Our previous research found that a mutation in *tpiA* promotes bacterial carbon metabolism, increasing the membrane potential and enhancing the uptake of aminoglycoside antibiotics [40]. In this study, we report that the mutation in *tpiA* leads to AgtR-mediated upregulation of CspC and subsequent translational repression of *rsaL*, resulting in increased expression of *lasI*. We further demonstrate that the AgtR-CspC-RsaL-LasI pathway is involved in the regulation of pyocyanin production in response to *Staphylococcus aureus* and GlcNAc. Overall, our results reveal a regulatory pathway linking QS systems with bacterial metabolism in a polymicrobial environment.

## Results

### Mutation of *tpiA* enhances quorum sensing systems

In our previous transcriptomic analysis, we found that mutation of *tpiA* increased the expression of pyocyanin biosynthesis genes and the elastase gene *lasB* [40]. We thus investigate the role of TpiA in regulating these genes. Bacteria with the same initial concentration were grown in LB and collected at the same growth phase ($OD_{600}$). Since the Δ*tpiA* mutant has a growth defect [40], it took longer time for the strain to reach the same $OD_{600}$ of the wild type PA14 and the complemented strain. The culture time and bacterial concentrations at corresponding $OD_{600}$ were listed in S1 Table. RT-qPCR confirmed the upregulation of *phzA* and *lasB* in the Δ*tpiA* mutant (S1A and S1B Fig). Since these genes are regulated by the QS systems in *P. aeruginosa*, we assessed the expression levels of the signal synthetase genes, *lasI*, *rhlI*, and *pqsA* at both stationary and exponential phases. In the Δ*tpiA* mutant, the mRNA levels of these genes were elevated compared to the wild type PA14, and this upregulation was restored by complementation with the *tpiA* gene (Figs 1A and S2A).

To further verify the upregulation of the QS systems, we employed transcriptional fusions of the promoters of *lasI*, *rhlI*, and *pqsA* with the *lacZ* gene (S3 Table). Consistent with the RT-qPCR results, LacZ levels were higher in the Δ*tpiA* mutant (Figs 1B and S2B). Additionally, the Δ*tpiA* mutant produced greater amounts of the QS signals, 3-oxo-$C_{12}$-HSL, $C_4$-HSL, and PQS (Figs 1C and S2C). Collectively, these findings demonstrate an upregulation of the QS systems in the Δ*tpiA* mutant at both exponential and stationary phases.

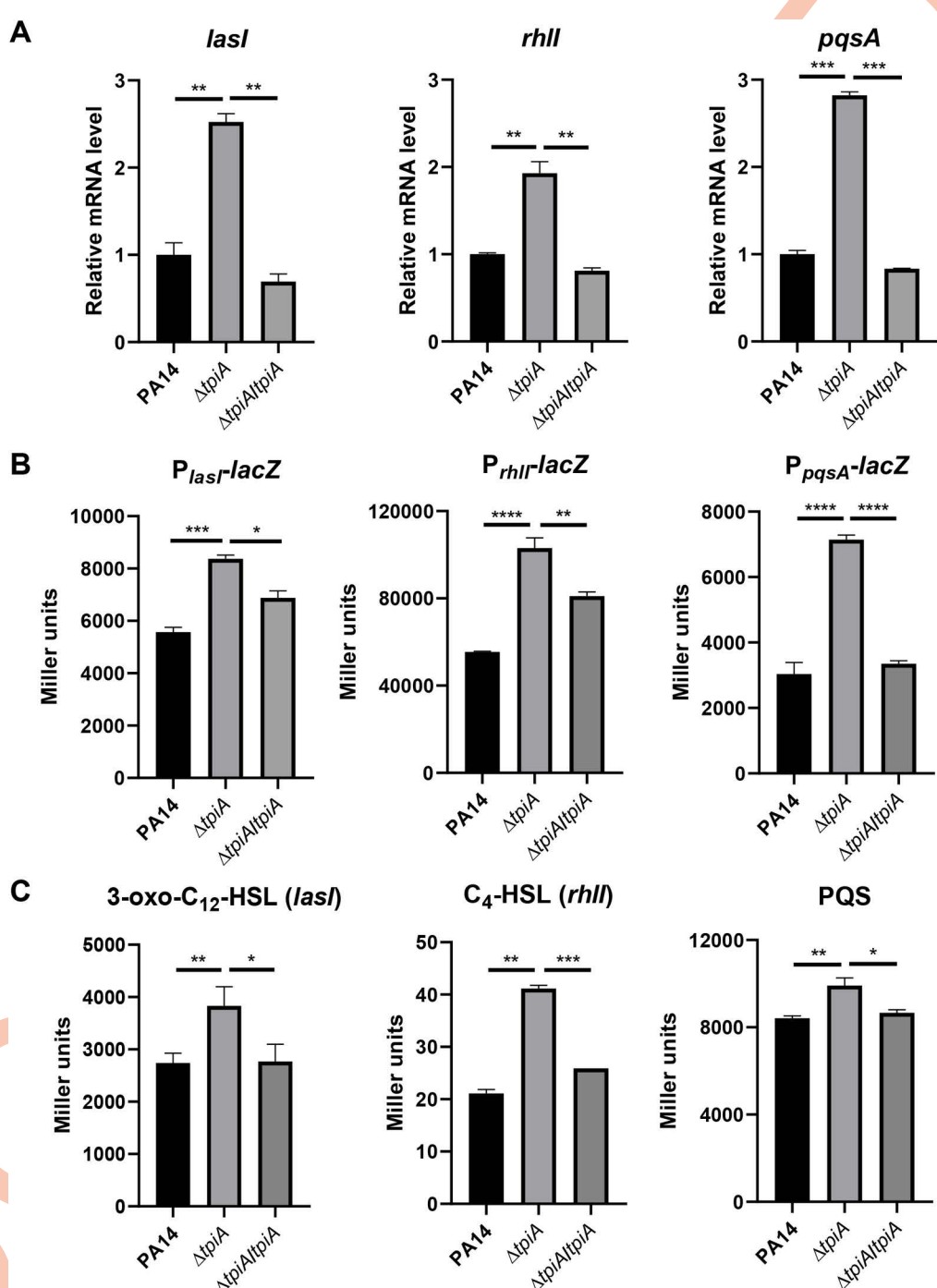

**Fig 1. Upregulation of the QS systems in the Δ*tpiA* mutant.** Bacteria were cultured in LB at 37 °C till OD$_{600}$ reached 2.5. (A) mRNA levels of the QS signaling molecule synthesis genes, including *lasI*, *rhlI* and *pqsA* were determined by RT-qPCR. (B) Promoter activities of *lasI*, *rhlI* and *pqsA* were detected by β-galactosidase activity assay. (C) The supernatants of indicated strains were collected. *E. coli* stains containing P$_{lasB}$–*lacZ*, P$_{rhlA}$-*lacZ* transcriptional fusions, and the PAO1 Δ*pqsA* mutant containing a P$_{pqsA}$-*lacZ* transcriptional fusion were incubated in the supernatant to measure the 3-Oxo-C$_{12}$-HSL, C$_4$-HSL and the PQS signal molecules (PQS and HHQ) levels using β-galactosidase activity assays, respectively. Data represent the mean ± standard deviation of the results from three samples. ****, $P < 0.0001$; ***, $P < 0.001$; **, $P < 0.01$; *, $P < 0.05$ by Student's *t* test.

## TpiA controls the Las system through RsaL

Given that the Las system is at the top of the QS regulatory hierarchy [18,19], we investigated whether TpiA influences the Las system. Beyond the positive feedback regulation of LasI-LasR, the *lasI* gene is negatively regulated by RsaL [20]. Using an *rsaL* promoter-*lacZ* transcriptional fusion (P*rsaL*-*lacZ*), we found that transcription of *rsaL* was upregulated in the Δ*tpiA* mutant (Fig 2A). RT-qPCR results further confirmed the enhanced mRNA levels of *rsaL* (Fig 2B). We then examined the translation of *rsaL* by constructing an *rsaL-gst* translational fusion driven by its native promoter (P*rsaL*-*rsaL-gst*). In contrast to the increased transcription of *rsaL*, the protein level of RsaL-GST was lower in the Δ*tpiA* mutant (Fig 2C). Using a pair of primers targeting the junction of the *rsaL* and *gst* coding regions, we verified that the mRNA level of *rsaL-gst* was higher in the Δ*tpiA* mutant (Fig 2D). Together, these results indicate that mutation of *tpiA* leads to translational repression of *rsaL*.

To investigate the mechanism underlying this translational repression, we deployed *rsaL-gst* translational fusions with varying lengths of the 5'-untranslated regions (5'-UTR) of *rsaL* driven by an exogenous *tac* promoter (Fig 3A) [39]. Previous studies have shown that *rsaL* transcription initiates 74 nucleotides (nt) upstream of its start codon [39]. Including the 74

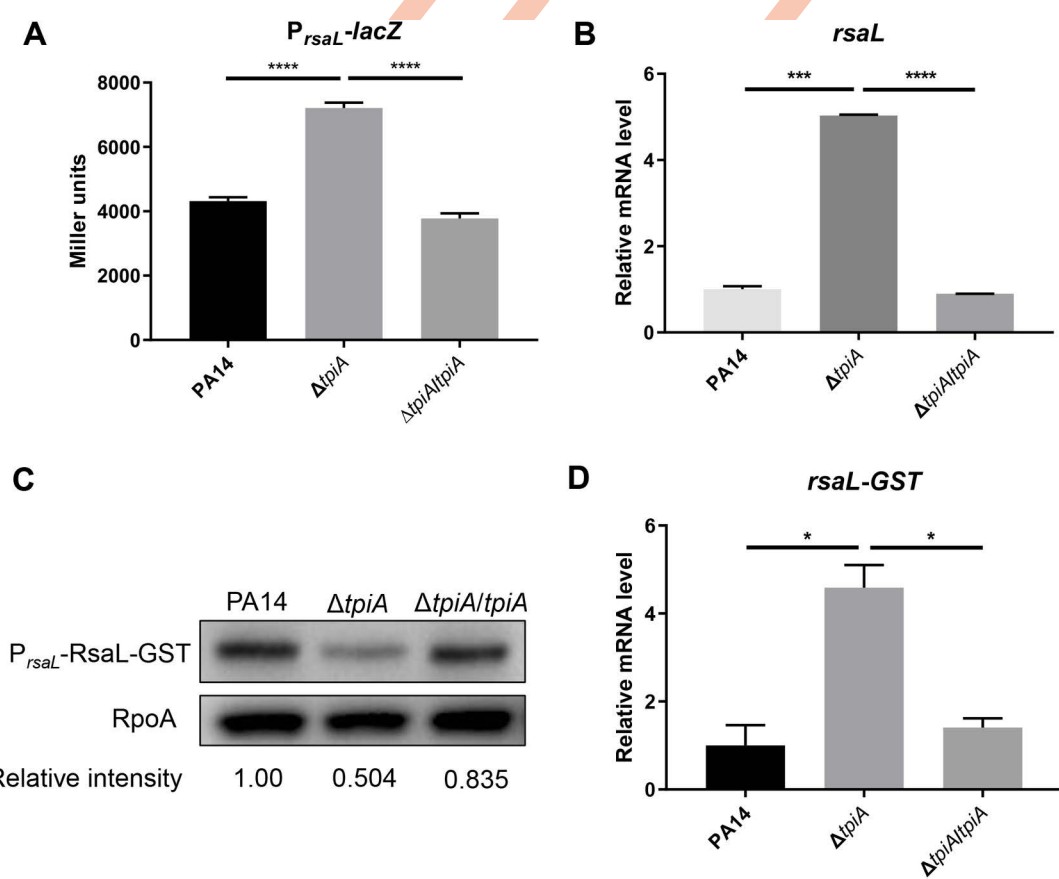

**Fig 2. TpiA controls the Las system through RsaL.** Bacteria were cultured in LB at 37 °C till OD$_{600}$ reached 2.5. (A) Promoter activities of *rsaL* were detected by β-galactosidase activity assay. **(B)** mRNA levels of *rsaL* were determined by RT-qPCR. Data represent the mean ± standard deviation of the results from three samples. (C) The RsaL-GST and RpoA levels were determined by Western Blot. **(D)** mRNA levels of *rsaL-gst* were determined by RT-qPCR. Data represent the mean ± standard deviation of the results from three samples. ****, $P < 0.0001$; ***, $P < 0.001$; *, $P < 0.05$ by Student's *t* test.

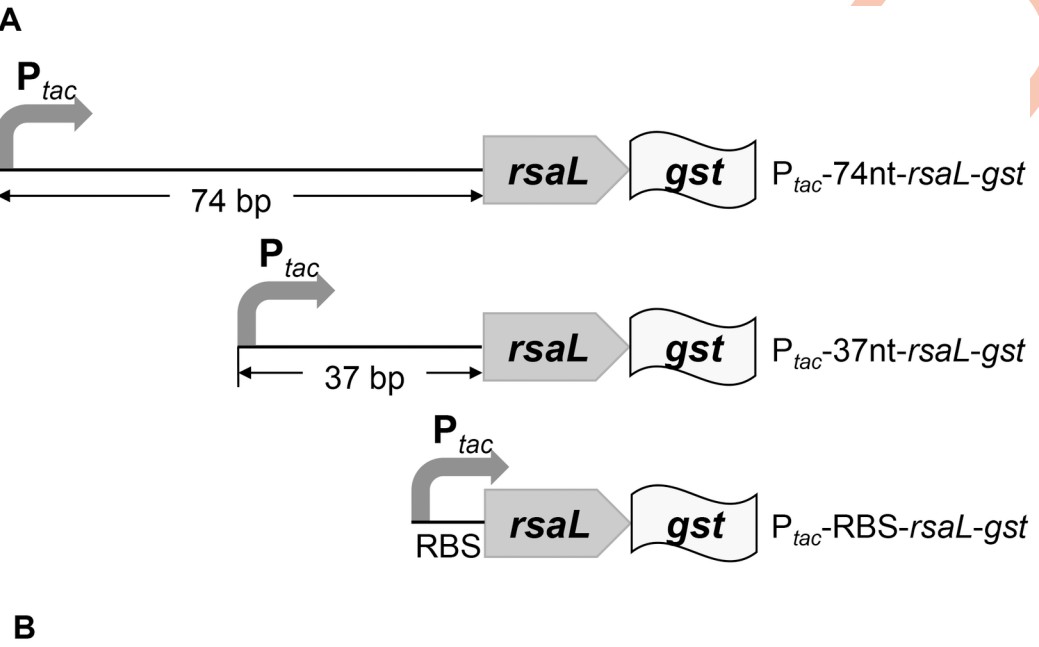

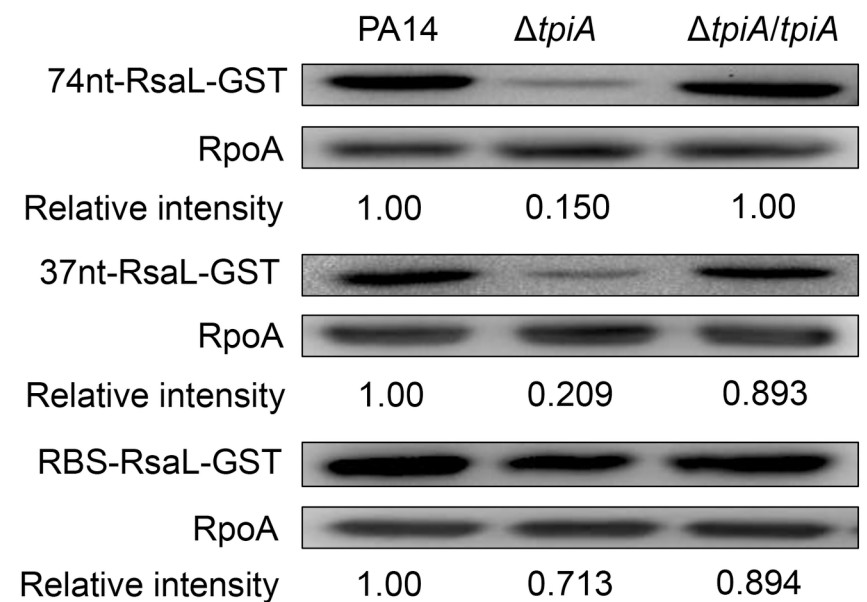

**Fig 3. Translational regulation of *rsaL*.** (A) Schematic diagram of the *rsaL-gst* fusions driven by a *tac* promoter with indicated 5'-UTRs. (B) Bacteria carrying the indicated *rsaL-gst* fusions were grown at 37 °C in LB containing 0.01 mM IPTG till OD$_{600}$ reached ~2.5. The RsaL-GST and RpoA levels were determined by Western Blot.

nt 5'-UTR (P$_{tac}$-74nt-*rsaL-gst*) resulted in lower amounts of RsaL-GST in the Δ*tpiA* mutant (Fig 3B). Reducing the 5'-UTR to 37 nt (P$_{tac}$-37nt-*rsaL-gst*) still resulted in a lower RsaL-GST level in the Δ*tpiA* mutant (Fig 3B). However, replacing the native 5'-UTR with an exogenous ribosome binding sequence (P$_{tac}$-RBS-*rsaL-gst*) led to similar RsaL-GST levels in both wild type PA14 and the Δ*tpiA* mutant (Fig 3B). These results indicate that the 5'-UTR of *rsaL* plays a role in the translational repression.

We then determined the role of RsaL in the upregulation of the Las system in the Δ*tpiA* mutant. Deletion of *rsaL* in wild type PA14 and the Δ*tpiA* mutant increased the mRNA level of *lasI* to a similar level (around 50-fold higher than PA14) (S3 Fig), demonstrating that TpiA regulates *lasI* through RsaL. The transcriptional level of *lasR* was upregulated 3-fold in the Δ*tpiA* mutant (S4 Fig). Deletion of *rsaL* in wild type PA14 increased the mRNA level of *lasR* by 2-fold. However, deletion of *rsaL* in the Δ*tpiA* mutant did not further increase the expression level of *lasR* (S3 Fig), indicating a different regulatory mechanism of *lasR* by RsaL.

## CspC regulates the translation of *rsaL* in the Δ*tpiA* mutant

We previously identified that the CspA family protein CspC represses the translation of *rsaL* mRNA by binding to its 37 nt 5'-UTR, and demonstrated that acetylation of CspC reduces its binding affinity [39]. We hypothesized that CspC might be upregulated or deacetylated in the Δ*tpiA* mutant. To assess the expression level of CspC, we utilized a *cspC-gst* translational fusion driven by its native promoter ($P_{cspC}$-*cspC-gst*). The level of CspC-GST was higher in the Δ*tpiA* mutant compared to wild type PA14 (Fig 4A). RT-qPCR results confirmed an increased mRNA level of *cspC* in the Δ*tpiA* mutant (Fig 4B). Using a transcriptional fusion of the *cspC* promoter and the *gst* gene ($P_{cspC}$-*gst*), we found enhanced promoter activity of *cspC* in the Δ*tpiA* mutant (Fig 4C). We then investigated the acetylation of CspC. CspC-GST was purified from the bacteria, revealing similar levels of acetylation in both wild type PA14 and the Δ*tpiA* mutant (S5 Fig).

To determine whether the upregulation of CspC contributes to the repression of *rsaL* in the Δ*tpiA* mutant, we constructed a Δ*tpiA*Δ*cspC* double mutant. The expression levels of *rsaL*, whether driven by its native promoter or the exogenous *tac* promoter, were increased in the Δ*tpiA*Δ*cspC* mutant to levels similar to those in wild type PA14 (Fig 5A and 5B). Concurrently, the expression of *lasI*, *rhlI*, *pqsA* and *phzA* in the Δ*tpiA* mutant was reduced upon deletion of *cspC* (Fig 5C). These results demonstrate that upregulation of *cspC* contributes to the repression of *rsaL* translation and the subsequent upregulation of the QS systems.

## AgtR upregulates *cspC* in the Δ*tpiA* mutant

To identify the regulator controlling the transcription of *cspC*, we performed a DNA pull-down assay using a biotin-conjugated DNA fragment containing the *cspC* promoter region (Biotin-$P_{cspC}$) and lysates from wild type PA14 and the Δ*tpiA* mutant. Compared to the negative control (no DNA probe), two distinct protein bands were pulled down by the Biotin-$P_{cspC}$ fragment (S6A Fig). Notably, these bands were more intense in samples from the Δ*tpiA* mutant than in those from wild type PA14 (S6A Fig). Mass spectrometric analysis of the protein bands revealed 12 transcriptional regulators (S4 Table). We then determined the expression levels of those genes by RT-qPCR. The mRNA levels of *gntR*, *PA0756* and *agtR* were elevated in the Δ*tpiA* mutant (S6B Fig).

To examine whether these genes regulate *cspC*, we firstly determined the expression levels of *cspC* in strains with mutations in those identified genes available from the PA14 transposon insertion mutant library [41]. Mutations in *gntR* and *agtR* reduced the expression level of *cspC* (S6C Fig). We then overexpressed *gntR* and *agtR* in wild type PA14 (PA14/pUCP20-*gntR* and PA14/pUCP20-*agtR*). RT-qPCR and β-galactosidase activity assays ($P_{cspC}$-*lacZ*) demonstrated that overexpression of *agtR* but not *gntR* resulted in upregulation of *cspC* (S6D and S6E Fig).

Furthermore, deletion of *agtR* in the Δ*tpiA* mutant decreased the activity of $P_{cspC}$ (Fig 6A) and the mRNA levels of *cspC*, *rsaL*, *lasI*, *phzA* and *agtA* (a gene known to be regulated by AgtR) (Fig 6B). Collectively, these results indicate that AgtR plays a role in upregulating *cspC* in the Δ*tpiA* mutant.

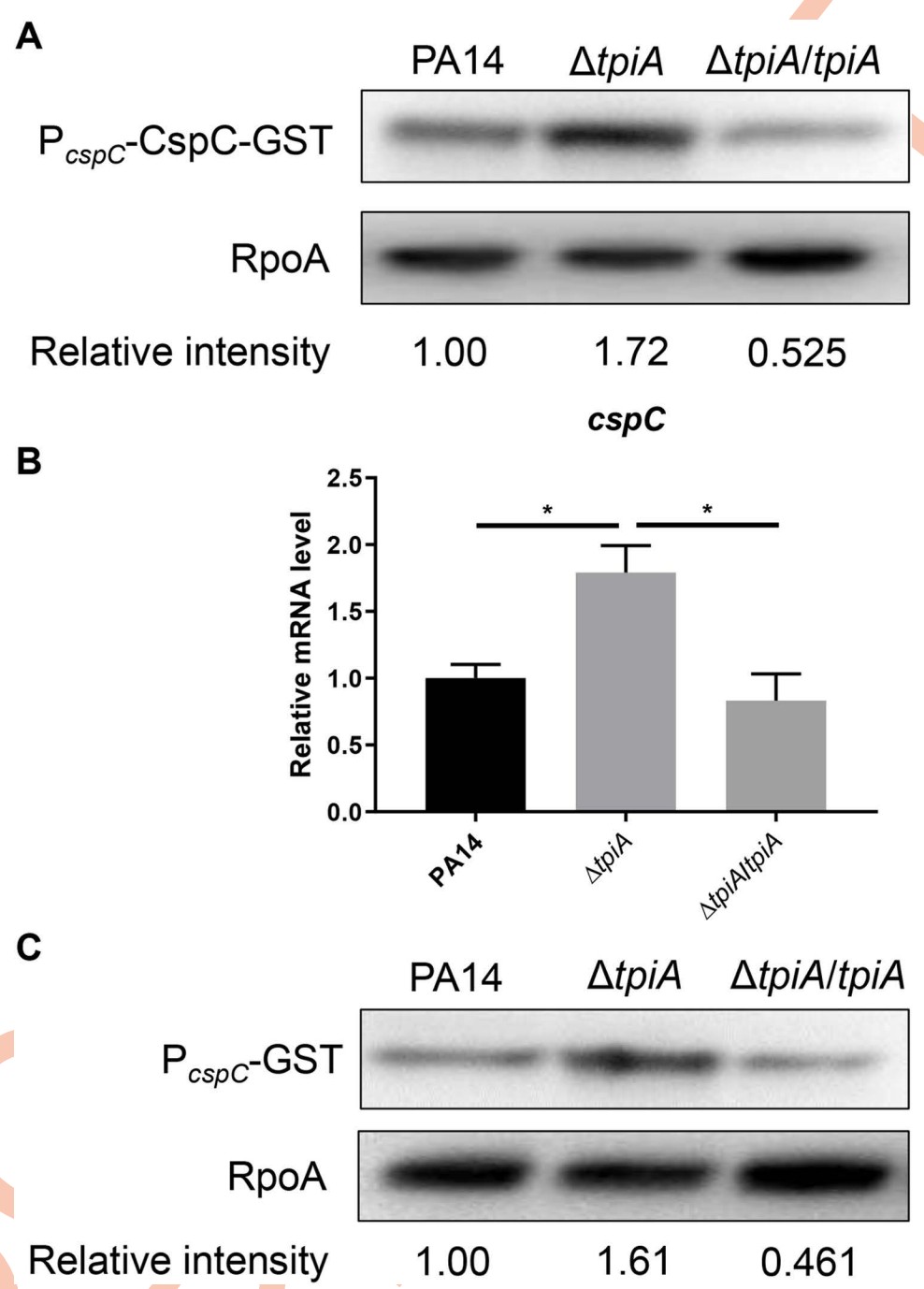

**Fig 4. The expression of *cspC* is upregulated in the Δ*tpiA* mutant.** (A) The CspC-GST and RpoA levels were determined by Western Blot. (**B**) mRNA levels of *cspC* were determined by RT-qPCR. (C) GST driven by P$_{cspC}$ was used for detecting the promoter activity of *cspC*. The GST and RpoA levels were determined by Western Blot. Data represent the means from three independent experiments, and error bars indicate standard deviations. *, $P < 0.05$ by Student's *t* test.

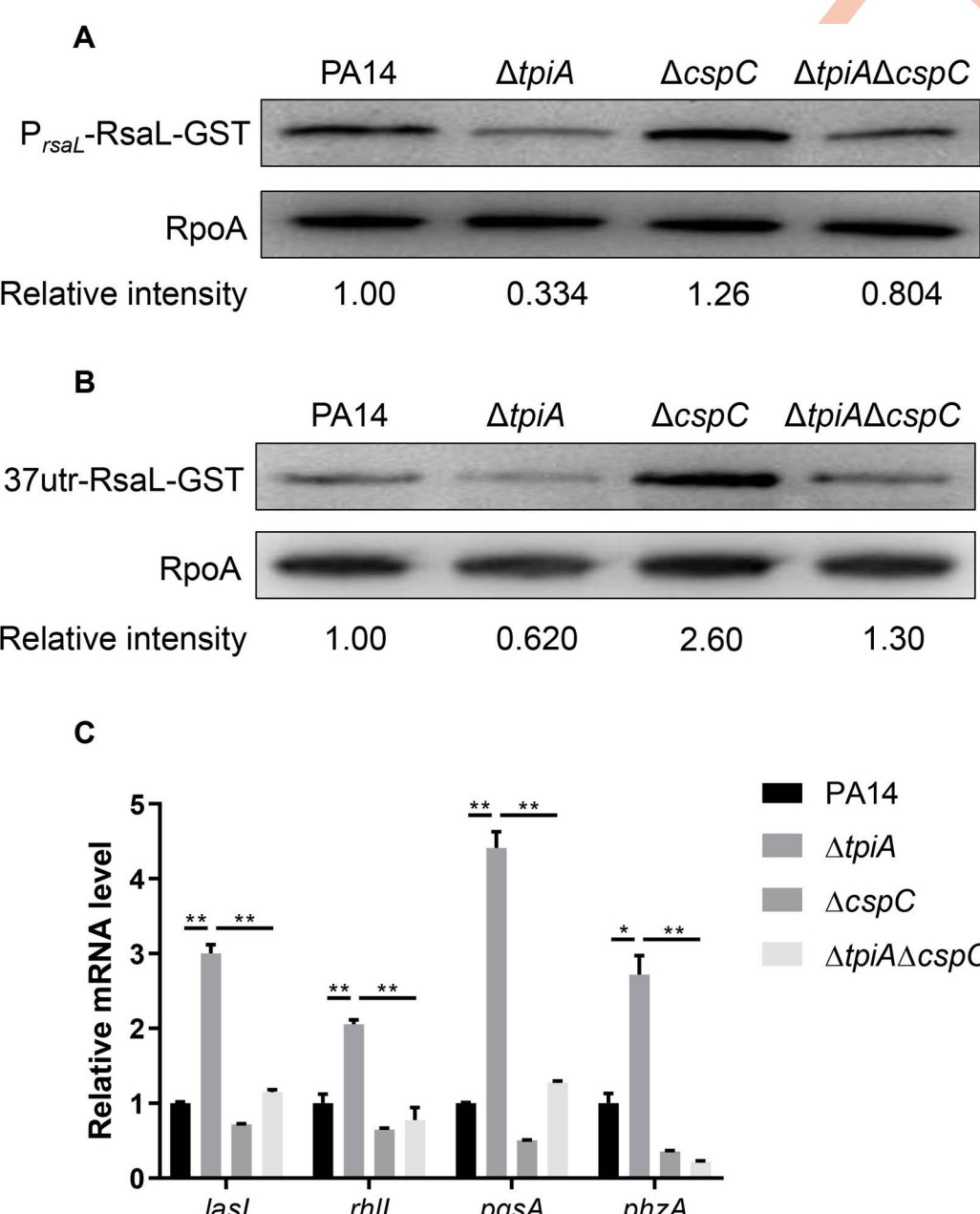

**Fig 5. CspC regulates the translation of *rsaL*.** (A) Bacteria carrying *rsaL-gst* driven by its native promoter were grown at 37 °C in LB till the OD$_{600}$ reached 2.5. The RsaL-GST and RpoA levels were determined by Western Blot. (B) Bacteria carrying P$_{tac}$-37-*rsaL-gst* were grown at 37 °C in LB containing 0.01 mM IPTG till the OD$_{600}$ reached 2.5. The RsaL-GST and RpoA levels were determined by Western Blot. **(C)** mRNA levels of *lasI*, *rhlI*, *pqsA* and *phzA* were determined by RT-qPCR. Data represent the mean ± standard deviation of the results from three samples. **, $P < 0.01$; *, $P < 0.05$ by Student's *t* test.

## AgtR directly regulates *cspC*

To determine whether AgtR directly activates the transcription of *cspC*, we conducted an electrophoretic mobility shift assay (EMSA) using purified His-SUMO-AgtR (S7 Fig) and a DNA fragment containing the *cspC* promoter region (Fig 7A). The DNA fragment was retarded

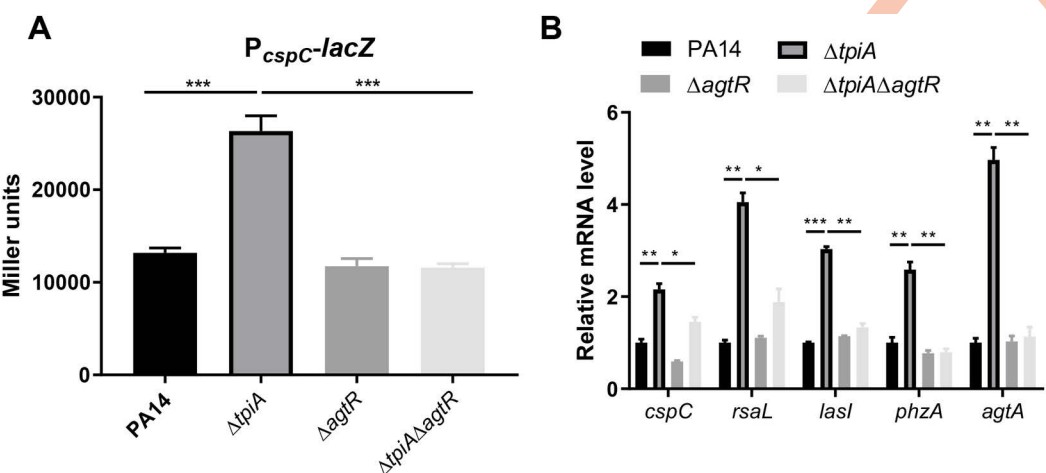

**Fig 6. AgtR upregulates *cspC* in the Δ*tpiA* mutant.** (A) Promoter activities of *cspC* were determined by β-galactosidase activity assay. (B) mRNA levels of *cspC*, *rsaL*, *lasI*, *phzA* and *agtA* were determined by RT-qPCR. Data represent the mean ± standard deviation of the results from three samples. \*\*\*, $P < 0.001$; \*\*, $P < 0.01$; \*, $P < 0.05$ by Student's *t* test.

by His-SUMO-AgtR (Fig 7B), demonstrating direct binding of AgtR to the *cspC* promoter region.

To identify the binding site of AgtR, we first located the transcription initiation site of *cspC* through a 5' rapid amplification of cDNA ends (5'-RACE) assay. As shown in S8 Fig, transcription of *cspC* initiates 106 bp upstream of its start codon. Based on this result, we performed EMSA with a series of probes with the length of 294-160 bp upstream of the *cspC* coding region (Fig 7A). The purified His-SUMO-AgtR was able to bind to probes ≥ 188 bp ($P_{cspC}$-1, $P_{cspC}$-2, $P_{cspC}$-3), but not to the 160 bp probe ($P_{cspC}$-4) (Fig 7C-7F). A fragment covering the -188 to -160 bp region ($P_{cspC}$-5) was bound by His-SUMO-AgtR (Fig 7G). A DNase I footprint assay located the potential AgtR binding site at 167-143 bp upstream of the *cspC* start codon (Fig 8A). Mutation of the binding sequence abolished the interaction between AgtR and the *cspC* promoter region (Fig 8B and 8C). Additionally, mutating the binding sequence in the $P_{cspC}$-*lacZ* transcriptional fusion reduced the *lacZ* expression in both the Δ*tpiA* mutant and the *agtR* overexpression strain to wild type levels (Fig 8D), demonstrating reduction of the *cspC* promoter activity. Collectively, these results demonstrate that AgtR directly activates the transcription of *cspC* by binding to its promoter region.

## The AgtR-CspC pathway is involved in sensing and competition against *S. aureus*

It has been shown that AgtR is necessary for sensing the peptidoglycan component GlcNAc and the subsequent activation of the QS systems, which play a crucial role in competition against *S. aureus* [28]. Based on our findings, we hypothesized that the AgtR-CspC pathway might be involved in the bacterial response to GlcNAc and in competition with *S. aureus*. Compared to glucose, GlcNAc induced the expression of *agtA*, *cspC*, *lasI*, *phzA*, *agtR* and pyocyanin production, while deletion of *agtR* abolished this induction (Fig 9A and 9B). In both M9 minimal and LB media, the presence of a *S. aureus* strain RN4220 induced the expression of these genes and pyocyanin production, which was eliminated by the deletion of *agtR* (Fig 9C-9F). Deletion of *cspC* also negated the GlcNAc or *S. aureus* RN4220-induced upregulation of *lasI* and *phzA* (Fig 9G-9I). We then examined the role of the AgtR-CspC pathway in the competition against *S. aureus*. Compared to wild type PA14, the Δ*agtR* and Δ*cspC* mutants

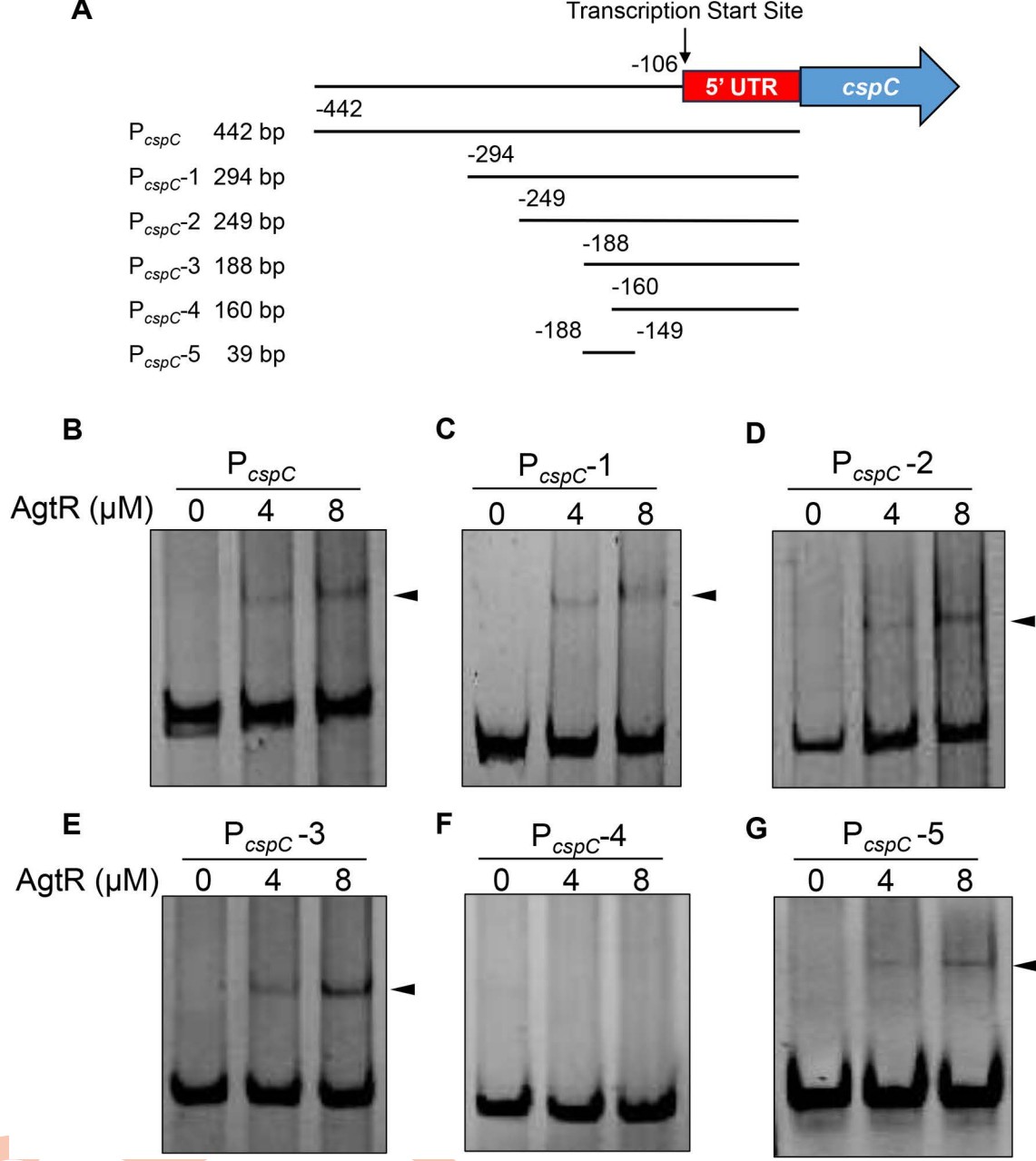

**Fig 7. AgtR binds to *cspC* promoter region.** (A) Diagram of DNA probes used in the EMSAs. The number indicates the position relative to the start codon of the *cspC* gene. (B-G) Indicated DNA probes (20 ng) were incubated with 0, 4 or 8 µM AgtR at room temperature for 30 min. The shifted band is indicated by arrowheads.

displayed defective competitive abilities against *S. aureus* RN4220 (Figs 9J and S9). To further confirm that pyocyanin plays an important role in the competition against *S. aureus*, we performed the competition assay with a Δ*phz1* mutant and a Δ*agtR*Δ*phz1* mutant, which are defective in pyocyanin production. The two strains also displayed impaired competitive abilities against *S. aureus* RN4220 (S9 Fig). Collectively, these results elucidate the roles of AgtR and CspC in sensing and competing against *S. aureus* by regulating pyocyanin production.

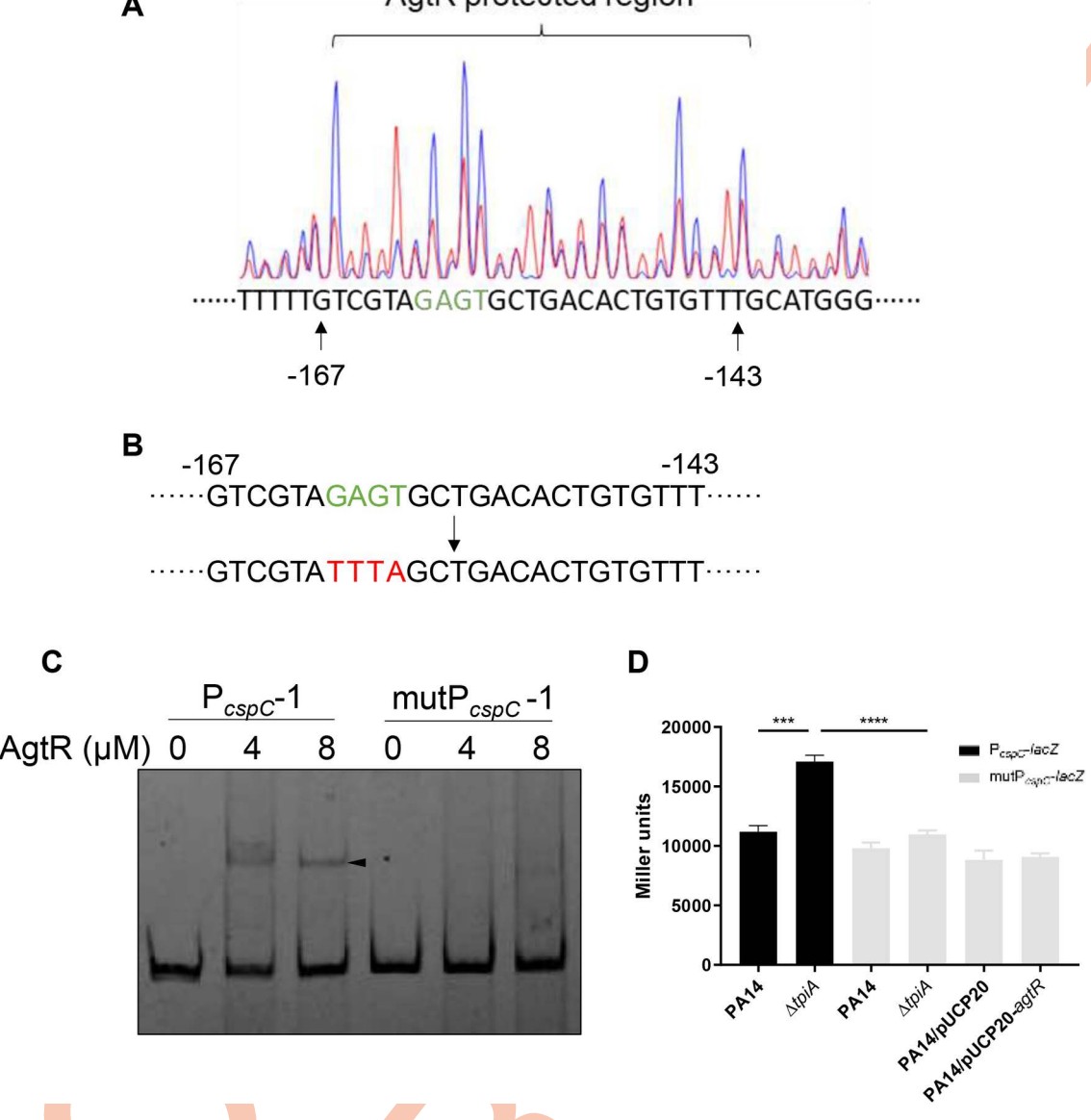

**Fig 8. AgtR directly regulates *cspC*.** (A) The potential AgtR binding sequence was determined by DNase I footprinting assay. Electropherograms were superimposed to show the region of P$_{cspC}$ protected by AgtR (red) compared with BSA (blue). The numbers indicate the position relative to the start codon of *cspC*. (B) The native sequence (shown in green) was mutated (shown in red). The numbers indicate the position (bp) upstream of the start codon. (C) The DNA probe and the mutated fragment (mutP$_{cspC}$ -1) were incubated with AgtR at room temperature for 30 min. The shifted band is indicated by an arrowhead. (D) Promoter activities of *cspC* and the one with the mutated sequence (mutP$_{cspC}$-*lacZ*) were determined by the β-galactosidase activity assay. Data represent the mean ± standard deviation of the results from three samples. ****, $P < 0.0001$; ***, $P < 0.001$ by Student's *t* test.

We further examined the role of the AgtR-CspC pathway in the competition of *P. aeruginosa* against other Gram-positive bacteria. Deletion of *agtR*, *cspC*, *phz1* or both *agtR* and *phz1* impaired the competitive abilities against *Streptococcus agalactiae* and *Streptococcus epidermidis* (S10 Fig), demonstrating the role of AgtR-CspC in competing against the two Gram-positive bacteria by regulating pyocyanin production.

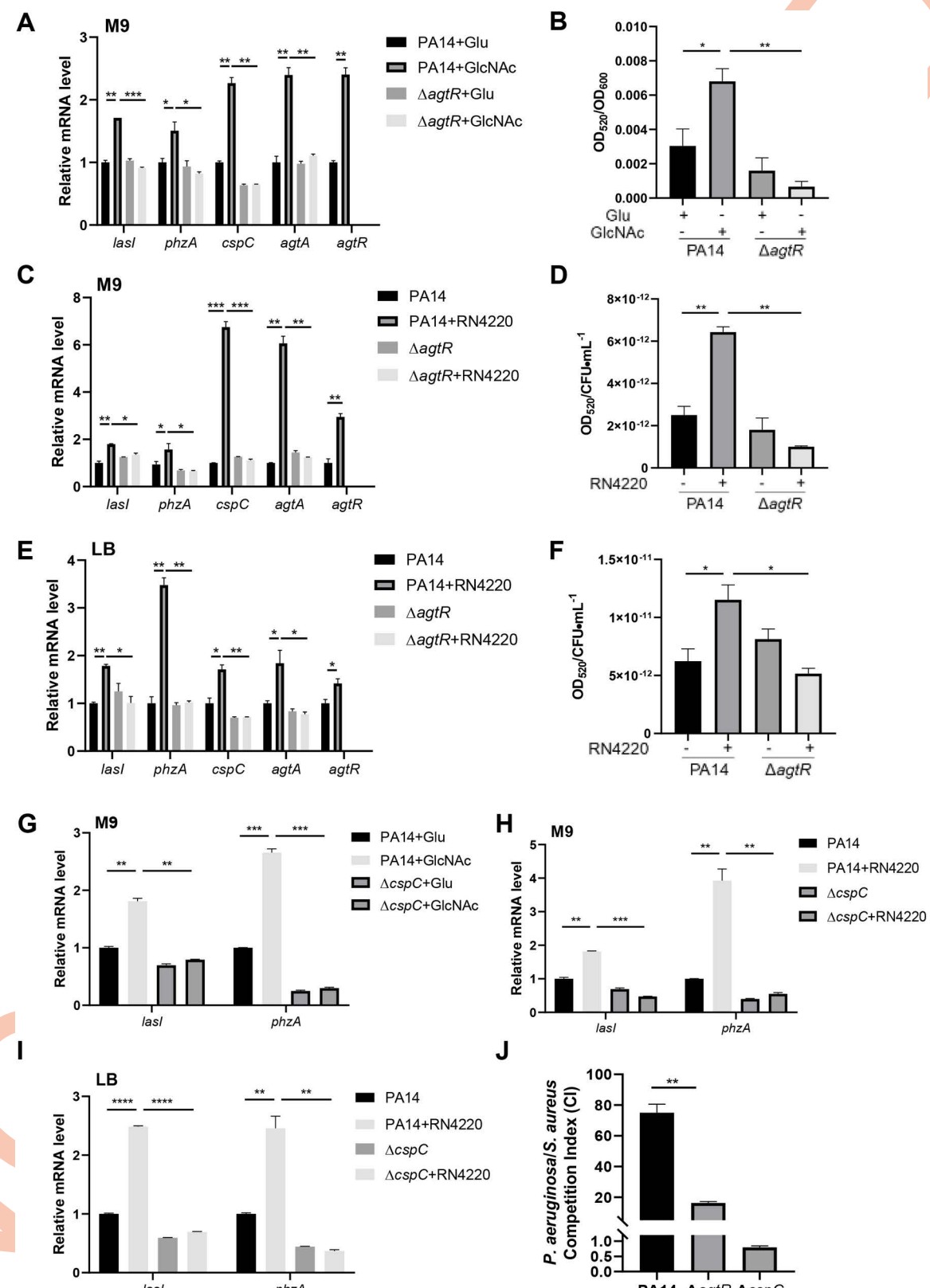

**Fig 9. AgtR and CspC are involved in bacterial responding to GlcNAc and competition against *S. aureus.*** (A-D) PA14 and the
Δ*agtR* mutant were cultured in the M9 medium with the addition of 2 mM Glu, GlcNAc or a *S. aureus* strain RN4220 (at the ratio of

1:3) at 37 °C till $OD_{600}$ reached 2.5, followed by determination of the levels of the *lasI*, *phzA*, *cspC*, *agtA* and *agtR* mRNA **(A, C)**, and pyocyanin in the supernatant (B, D). PA14 and the Δ*agtR* mutant were cultured in LB with the addition of *S. aureus* RN4220 (at the ratio of 1:10) at 37 °C till $OD_{600}$ reached 2.5, followed by determination of the levels of the *lasI*, *phzA*, *cspC*, *agtA* and *agtR* mRNA (E), and pyocyanin in the supernatant (F). PA14 and the Δ*cspC* mutant were cultured in the M9 medium with the addition of 2 mM Glu, GlcNAc or *S. aureus* RN4220 (at the ratio of 1:3), or in LB with or without *S. aureus* RN4220 (at the ratio of 1:10) at 37 °C till $OD_{600}$ reached 2.5, followed by determination of the levels of the *lasI*, *phzA* mRNA (G-I). (J) *P. aeruginosa* strains and *S. aureus* RN4220 were mixed at a 1:30 ratio and cocultured in LB at 37 °C for 14 h. The competitive indexes represent the ratio of *P. aeruginosa* to *S. aureus*. Data represent the mean ± standard deviation of the results from three samples. ***, $P < 0.001$; **, $P < 0.01$; *, $P < 0.05$ by Student's *t* test.

## Discussion

The QS systems of *P. aeruginosa* play critical roles in bacterial virulence and interspecies competition [11–13]. Various environmental stimuli and metabolic processes have been shown to influence these QS systems [21,22]. In this study, we identified an AgtR-CspC-RsaL-LasI regulatory pathway that modulates the QS in response to metabolic alternations and extracellular GlcNAc, as illustrated in Fig 10.

*P. aeruginosa* possesses a complex metabolic network, allowing it to utilize diverse carbon sources. Due to the absence of phosphofructokinase, this organism cannot convert fructose-6-phosphate into fructose-1,6-biphosphate [42]. Instead, glucose is processed through the Entner-Doudoroff (ED) pathway, yielding glycerol-3-phosphate (G3P) and dihydroxyacetone phosphate (DHAP) [43]. The conversion between DHAP and G3P catalyzed by TpiA is a crucial link between glycerol metabolism and glycolysis/gluconeogenesis [42]. Previous research shows that *tpiA* mutation boosts respiratory activity, intracellular NADH levels, and membrane potential. These effects can be reduced by the TCA cycle inhibitor $Na_2ATP$ [40]. Thus, the *tpiA* mutation likely boosts bacterial respiration by increasing carbon flow into the TCA cycle.

In this study, we found that *tpiA* mutation leads to AgtR-mediated upregulation of *agtA*. Along with *agtB*, *agtC*, and *agtD*, *agtA* forms an operon responsible for transporting γ-aminobutyrate and δ-aminovalerate. This operon is regulated by the AgtS-AgtR two-component regulatory system in response to substrate availability [29]. Notably, Korgaonkar *et al.* demonstrated AgtR's role in regulating pyocyanin production in response to GlcNAc [28], suggesting that GlcNAc may serve as a ligand for the AgtS-AgtR system. Given TpiA's role in metabolism, it is plausible that the *tpiA* mutant overproduces AgtS-AgtR ligands. Ongoing metabolomic analyses aim to elucidate the altered metabolic pathways in the *tpiA* mutant.

It has been demonstrated that AgtS-AgtR can sense δ-aminovalerate, γ-aminobutyrate and β-alanine [29], as well GlcNAc [28]. All four compounds contain an amino group ($-NH_2$). δ-aminovalerate, γ-aminobutyrate, and β-alanine are amino acids with a carboxyl group ($-COOH$), while GlcNAc has an acetylamino group ($-CONH_2$) instead of a free amino group. δ-aminovalerate, γ-aminobutyrate, and β-alanine contain a five-, four- and three-carbon chain, respectively. GlcNAc, on the other hand, is a larger molecule with an eight-carbon glucose backbone. Since all of the compounds are able to activate the AgtS-AgtR pathway, it is likely that they are ligands of AgtS. Further studies are warranted to examine the direct interaction between AgtS and each of the compounds.

In addition to the *agtABCD* operon, we confirmed that *cspC* is directly regulated by AgtR in response to GlcNAc. CspC, a member of the CspA family, binds mRNAs to regulate their translation [44]. We previously demonstrated that CspC binds to *rsaL* mRNA, repressing its translation [39]. AgtR's regulation of pyocyanin production in response to GlcNAc, contributes to the competition against *S. aureus* in polymicrobial infections [28], highlighting the interconnectedness of metabolic signaling and virulence in *P. aeruginosa*. Our findings fill

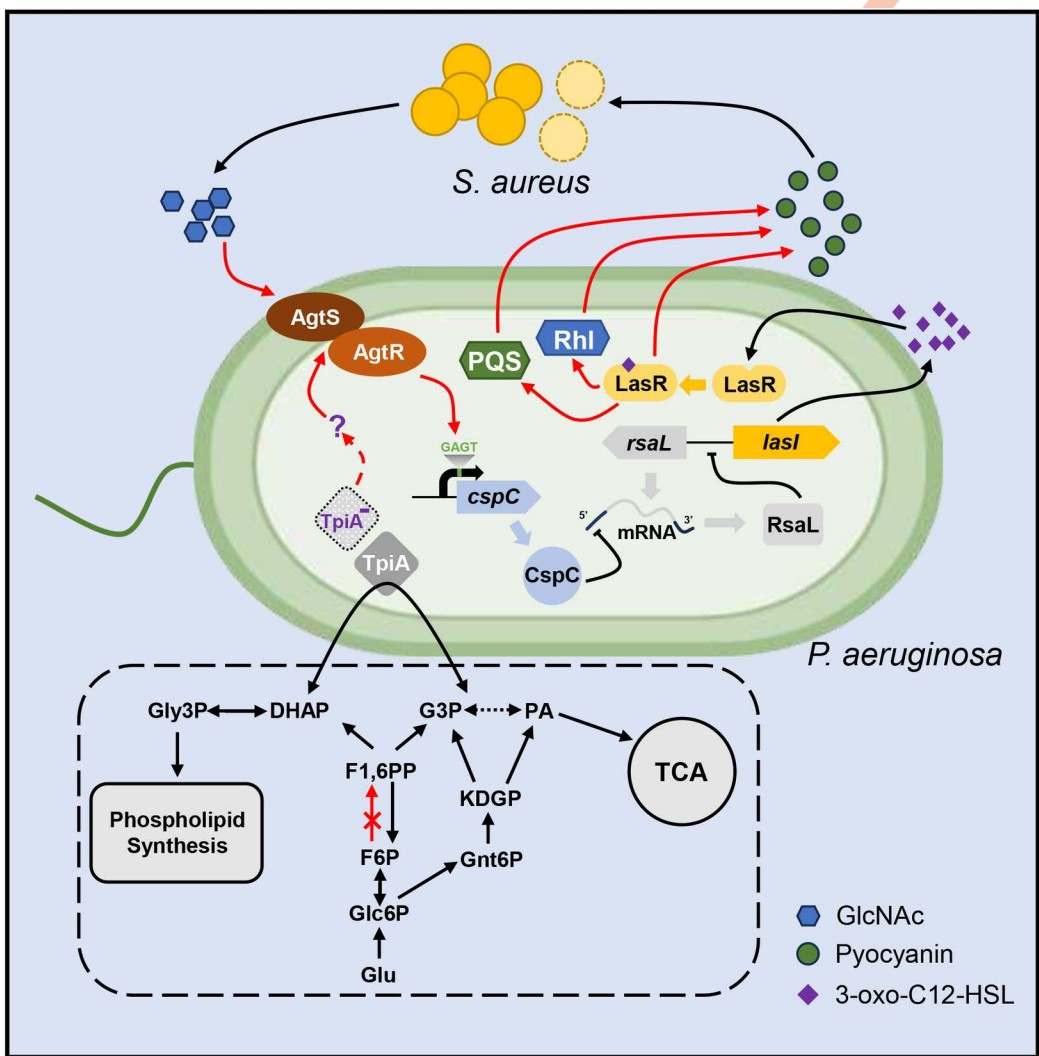

**Fig 10. Schematic diagram of the AgtR-CspC-RsaL-LasI pathway.** *P. aeruginosa* lacks the phosphofructokinase that converts fructose-6-phosphate (F6P) into fructose-1,6-biphosphate (F1,6PP). Glyceraldehyde-3-phosphate (G3P) and pyruvate (PA) are derived from the ED pathway. TpiA catalyzes the reversible conversion between G3P and dihydroace-tone phosphate (DHAP), which links glycolysis/gluconeogenesis, the TCA cycle and glycerol metabolism/phospholipid synthesis. Through an unknown mechanism, mutation of *tpiA* (TpiA⁻) activates AgtR, which increases the transcription of *cspC*, leading to repression of *rsaL* translation and subsequent upregulation of *lasI* and *lasR*. The Las system activates the Rhl and PQS systems, which positively regulate pyocyanin production. In addition, since RsaL directly represses the transcription of *phzA1*, and *phzM* [60], repression of the *rsaL* translation derepresses the expression of the two genes. The *S. aureus* released GlcNAc is presumably sensed by AgtS, leading to the activation of the AgtR-CspC-RsaL-LasI pathway and subsequent production of pyocyanin, which is toxic to *S. aureus*. Glu, glucose; Glc6P, glucose-6-phosphate; Gnt6P, 6-phosphogluconate; Gly3P, glycerol-3-phosphate; KDPG, 2-keto-3-deoxy-6-phosphogluconate. The dotted line indicates multiple reactions, and the dashed line indicates an unknown mechanism.

critical gaps in this regulatory pathway, reinforcing the complex relationship between metabolism and QS.

Moreover, we previously observed that itaconate reduces CspC acetylation, thereby enhancing its affinity for *rsaL* mRNA and subsequently upregulating *lasI* [39]. Itaconate, an immunometabolite produced by macrophages during infection, exhibits anti-inflammatory properties [45–47]. *P. aeruginosa* encodes itaconate CoA transferase (PA0882), itaconyl-CoA

hydratase (PA0878), and (S)-citramalyl-CoA lyase (PA0883), which convert itaconate into pyruvate and acetyl-CoA [48]. Additionally, itaconate serves as a signaling molecule that reduces lipopolysaccharide (LPS) synthesis while increasing extracellular polysaccharide (EPS) production [49]. Together, these findings suggest that in response to itaconate, *P. aeruginosa* may mitigate host inflammation by reducing LPS synthesis while enhancing biofilm formation through the activation of QS and EPS production, thereby promoting persistence in the host.

Overall, our results elucidate the AgtR-CspC-RsaL-LasI pathway, demonstrating its regulatory role in the QS systems in response to metabolite alternation. This pathway contributes significantly to bacterial pathogenesis and interspecies competition, emphasizing the intricate connections between metabolism, signaling, and virulence in *P. aeruginosa*.

## Materials and methods

### Bacterial strains and plasmids

The primers, plasmids, and bacterial strains used in this study are listed in S2 and S3 Tables. A comprehensive description of plasmid construct methods is provided in S1 Text. Bacteria were cultured in LB or M9 minimal medium at 37 °C, with or without indicated antibiotics. Antibiotics were purchased from BBI Life Sciences, Shanghai, China, and the working concentrations are as follows: ampicillin, 100 µg/mL for *E. coli*; carbenicillin, 150 µg/mL for *P. aeruginosa*; gentamicin, 50 µg/mL for *P. aeruginosa*; kanamycin, 50 µg/mL for *E. coli*; and tetracycline, 10 µg/mL for *E. coli* and 50 µg/mL for *P. aeruginosa*.

### RNA isolation and quantitative real-time PCR

Bacteria were harvested at the exponential ($OD_{600} \approx 0.5$) or stationary phase ($OD_{600} \approx 2.5$). Total RNA was isolated using the Bacteria Total RNA Kit (Zomanbia, Beijing, China), and cDNAs were synthesized using reverse transcriptase (TaKaRa, Dalian, China). Quantitative real-time PCR was conducted with the ChamQ Universal SYBR qPCR Master Mix (Vazyme, Nanjing, China), using the *rpsL* gene, which encodes the 30 S ribosomal protein, as an internal control [50,51]. Data were analyzed using the CFX Connect Real-Time system (Bio-Rad, USA) with the $2^{-\Delta\Delta CT}$ method.

### *β*-galactosidase activity assay

The *β*-galactosidase activity assay was conducted as previously described [52], with minor modifications. Bacteria were cultured in LB medium at 37 °C until reaching an $OD_{600}$ of approximately 0.5 or 2.5. Two milliliters of culture were harvested and resuspended in 2 mL Z buffer (0.06 M $Na_2HPO_4$, 0.04 M $NaH_2PO_4$, 0.01 M KCl, 0.001 M $MgSO_4$, and 0.05 M β-mercaptoethanol, pH 7.0). A 500 µL aliquot of this suspension was mixed with 10 µL of 0.1% sodium dodecyl sulfate (SDS) and 10 µL of chloroform, then vortexed. After adding 100 µL of *o*-nitrophenyl-*β*-D-galactoside (ONPG, 4 mg/mL in Z buffer), the mixture was incubated at 37 °C. Once the mixture turned light yellow, 500 µL of 1 M $Na_2CO_3$ was added to stop the reaction. The $OD_{420}$ of the reaction mixture was then measured, while another 500 µL of the suspension was used for $OD_{600}$ measurement. *β*-galactosidase activities (Miller units) were calculated using the formula: Miller units = $(1000 \times OD_{420})/(500 \times T \times OD_{600})$; where T is the reaction time in minutes.

### QS signal molecular reporter assay

The QS signal molecular reporter assay was performed as previously described [53]. *E. coli*-QS signal reporter strains were utilized to assess the relative levels of 3-oxo-$C_{12}$-HSL and $C_4$-HSL,

while a *P. aeruginosa* QS signal reporter strain was used for PQS determination. Wild type PA14, the ΔtpiA mutant, and complemented strains were cultured in LB medium at 37 °C until $OD_{600}$ reached ~0.5 or 2.5. Supernatants were collected by centrifugation.

For the *E. coli* reporter strains, 1 mL of supernatant was mixed with 3 mL of DH5α ($OD_{600}$ 0.1) containing either pECP64 (reporter plasmid for 3-oxo-$C_{12}$-HSL, $P_{lasB}$-*lacZ*) or pECP61.5 (reporter plasmid for $C_4$-HSL, $P_{rhlA}$-*lacZ*) [53]. When the $OD_{600}$ reached 0.3~0.4, 1 mM isopropyl-*β*-D-thiogalactopyranoside (IPTG) was added. After 1 hour, the *β*-galactosidase activities were measured as previously described [53].

For the PQS system signal molecules (PQS and HHQ) determination, the *P. aeruginosa* reporter strain (PAO1 ΔpqsA mutant) containing a transcriptional fusion of the *pqsA* promoter and the *lacZ* gene ($P_{pqsA}$-*lacZ*) on a pDN19*lacZ*Ω plasmid [34] was cultured in LB medium at 37 °C until $OD_{600}$ reached ~1.0. One milliliter of the reporter bacteria was harvested by centrifugation. Then the bacteria were resuspend in 1 mL of supernatant from the tested strains, followed by a 30-minute incubation at room temperature before measuring *β*-galactosidase activities.

## Western blot

Protein samples from equal amounts of bacteria were collected by centrifugation, resuspended in 1 ×SDS sample buffer [62.5 mM Tris-HCl (pH 6.8), 2% (w/v) SDS, 0.02% (w/v) bromophenol blue, 1% (v/v) β-mercaptoethanol and 10% (v/v) glycerol], and boiled for 10 min. The samples were then separated by SDS-polyacrylamide (12%) gel electrophoresis (SDS-PAGE) and transferred onto a polyvinylidene difluoride (PVDF) membrane. The membrane was blocked with 5% nonfat milk in PBST [PBS with 0.1% (v/v) Tween 20] for at least 1 h at room temperature, followed by a 2-hour incubation with a rabbit anti-Glutathione-S-Transferase (GST) tag polyclonal antibody (Applygen, China) or a mouse monoclonal anti-RNA polymerase α antibody (Biolegend) at room temperature. After washing with PBST four times for 7 min each, the membrane was incubated with corresponding secondary antibodies at room temperature for 1 h. The membrane was washed again four times with PBST, and the signals were detected using an Immobilon Western kit (Millipore).

## Determination of CspC acetylation

The *cspC*-*gst* fusion driven by the native *cspC* promoter on pUCP20 ($P_{cspC}$-*cspC*-*gst*) was introduced into wild type PA14, the ΔtpiA mutant, and the complemented strain via electroporation. Bacteria were grown in LB to an $OD_{600}$ of ~1.0 and collected by centrifugation. Purification of the CspC-GST protein was conducted using a GST-tag Protein Purification kit (Beyotime, China). For determination of acetylation, equivalent amounts of the purified proteins were separated by a 12% SDS-PAGE gel, and then transferred onto a PVDF membrane. The membrane was incubated with either a rabbit anti-GST tag polyclonal antibody (Applygen, China) or a mouse monoclonal anti-acetyllysine antibody (Jingjie PTM Biolab, China). Signals were detected with an Immobilon Western kit (Millipore).

## DNA pull-down assay

The DNA pull-down assay was performed as previously described, with minor modifications [54,55]. Streptavidin magnetic beads (Streptavidin Mag Sepharose, Cytiva) were washed three times with B/W buffer (10 mM Tris, 1 mM EDTA, 2 M NaCl, pH 7.5), and then resuspended in 200 μL of B/W buffer. The 5'-biotinylated DNA fragment of the $P_{cspC}$ promoter was amplified by PCR using the PA14 genome as a template with primers, PcspC-F-Biotin and PcspC-R (S2 Table). The amplified DNA fragment was purified and incubated with the

washed streptavidin magnetic beads at room temperature for 1 h. The beads were then washed three times with BS/THES buffer, which is composed of a BS buffer (10 mM HEPES, 5 mM $CaCl_2$, 50 mM KCl, 60% (v/v) glycerol) and a THES buffer [50 mM Tris, 10 mM EDTA, 20% (w/v) sucrose, 140 mM NaCl 0.7% (v/v) Protease Inhibitor Cocktail II (Sigma), 0.1% (v/v) Phosphatase Inhibitor Cocktail II (Sigma), pH 7.5] (at an 1:1 ratio). Two hundred milliliters of PA14 or the $\Delta tpiA$ mutant were grown to the stationary phase ($OD_{600} \approx 2.5$), then harvested by centrifugation at $8000 \times g$ for 10 min at 4 °C, resuspended in 4 mL BS/THES buffer, and lysed by sonication on ice. After centrifugation at $12000 \times g$ for 10 min at 4 °C, 2 mL of the supernatant was mixed with the immobilized DNA fragments on the magnetic beads and incubated at room temperature for 30 min. Following from that, the beads were harvested, mixed with another 2 mL of the bacterial supernatant and incubated at 4 °C for 1 h. Next, the beads were washed five times with 500 μL BS/THES buffer to remove nonspecifically bound proteins, and the DNA binding proteins were eluted by washing with 1 M NaCl. The negative control was cell lysates from PA14 and the $\Delta tpiA$ mutant incubated with magnetic beads without DNA. The eluted proteins were separated by SDS-PAGE and stained with Coomassie blue. The protein bands of interest (found in samples from the DNA-coupled beads, but absent in the negative control) were excised from the SDS-PAGE gel and analyzed via mass spectrometry.

## Protein purification

The *E. coli* strain BL21 (DE3) containing pET-His-SUMO-*agtR* was cultured in 150 mL LB medium at 37 °C to an $OD_{600}$ of 0.4-0.6, followed by addition of 0.5 mM IPTG. After grown overnight (~12-16 h) at 20 °C, the *E. coli* cells were harvested by centrifugation at $8000 \times g$ for 10 min at 4 °C, then resuspended in the lysis buffer (50 mM sodium phosphate, 0.3 M NaCl, pH 8.0) and lysed by sonication on ice. After centrifugation at $12000 \times g$ for 10 min at 4 °C, the supernatant was incubated with Ni-NTA resin (Qiagen) at 4 °C for 2 h. The Ni-NTA resin was subsequently washed with the lysis buffer containing 30 mM imidazole to remove nonspecific binding proteins. Next, His-SUMO-AgtR was eluted with lysis buffer containing 300 mM imidazole. The purified protein was examined by SDS-PAGE and stained with Coomassie blue.

## Electrophoretic mobility shift assay (EMSA)

EMSA was performed as previously described with minor modifications [56]. DNA fragments were amplified by PCR using the PA14 genome as the template with specific primers (S2 Table). Twenty nanograms of DNA probes were incubated with increasing amounts of the purified His-SUMO-AgtR protein at room temperature for 30 min in a 20 μL reaction mixture containing 50 mM Tris-HCl (pH 8.0), 50 mM KCl, 1 mM EDTA and 5% (v/v) glycerol. The samples were loaded onto an 8% or 12% native polyacrylamide gel in 0.5 × TBE (Tris-borate-EDTA) buffer (44.5 mM Tris base, 44.5 mM boric acid, 1 mM EDTA, pH 8.0) and electrophoresed on ice at 10 mA for 50, 70 or 90 min. The gel was stained with ethidium bromide and imaged using a molecular imager ChemiDoc XRS+ (Bio-Rad, CA, USA).

## 5' rapid amplification of cDNA ends (5'-RACE) assay

Total RNA (1 μg) purified from PA14 and a *cspC*-specific primer (S2 Table) were used to synthesize the 5'-RACE-Ready first-strand cDNA template using a PrimeScriptRT reagent kit (TaKaRa, Dalian, China). After adding poly dG to the 5'-terminal of cDNA using a Terminal Deoxynucleotidyl Transferase Kit (TaKaRa, Dalian, China), PCR was performed with a universal forward primer and the *cspC*-specific reverse primer (S2 Table). The transcription start site was mapped by sequencing the PCR product.

## DNase I footprint assay

DNase I footprint assay was performed as previously described, with minor modification [57]. DNA fragments were amplified with specific primers, FAM-PcspC-F and PcspC-R (S2 Table). The forward primer was labelled with 6-carboxyfluorescein (6-FAM). The 6-FAM labelled PCR product (200 ng) was mixed with 4 or 8 μM AgtR, or 8 μM BSA. After adding 0.05 U of DNase I (TaKaRa, Dalian, China) into every reaction mixture, the samples were incubated at 25 °C for 5 min and then heated at 75 °C for 10 min to halt the digestion. The digested DNA was purified and analyzed by GENEWIZ (Suzhou, China). Sequences of the protected regions were analyzed by the Peak Scanner software v1.0 (Applied Biosystems, CA, USA).

## Pyocyanin production assay

The levels of pyocyanin production were determined as previously described [58]. Bacteria were cultured in LB or M9 minimal medium at 37 °C. 1.5 mL of the bacterial culture was centrifuged at $13000 \times g$ for 2 minutes. 1 mL of the supernatant was mixed with 600 μL chloroform. After centrifugation at $13000 \times g$ for 5 minutes, the lower layer (400 μL) was mixed with 300 μL 0.2 M HCl, then separated by centrifugation at $13000 \times g$ for 5 minutes. The upper layer was taken for the measurement of $OD_{520}$ by a microplate reader (Bio-Rad, USA).

## Competition assay

The *P. aeruginosa-S. aureus* competition assay was performed as previously described with minor modifications [59]. Bacteria were cultured overnight in LB at 37 °C. *P. aeruginosa* and *S. aureus* were mixed at a 1:30 ratio and cocultured in LB at 37 °C for 14h. *P. aeruginosa* and *S. agalactiae* or *S. epidermidis* were mixed at a 1:3 ratio or a 1:30 ratio, and cocultured in LB at 37 °C for 6h. Bacterial cultures were serially diluted and plated on selective media (*Pseudomonas* Isolation Broth for *P. aeruginosa,* Mannitol Salt Agar for *S. aureus* and LB medium containing 4 μg/mL colistin sulfate for *S. agalactiae* and *S. epidermidis*) to determine the CFUs of *P. aeruginosa*, *S. aureus*, *S. agalactiae* and *S. epidermidis*, respectively. The competitive index (CI) was calculated as the ratio of *P. aeruginosa* to *S. aureus*, *S. agalactiae* or *S. epidermidis* bacterial CFU.

## Supporting information

**S1 Fig. Upregulation of *phzA* and *lasB* in the Δ*tpiA* mutant.** Bacteria were cultured in LB at 37 °C till $OD_{600}$ reached 2.5. mRNA levels of *phzA* (A) and *lasB* (B) were determined by RT-qPCR. Data represent the mean ± standard deviation of the results from three samples. \*\*\*, $P < 0.001$; \*\*, $P < 0.01$ by Student's *t* test.
(TIF)

**S2 Fig. Upregulation of the QS systems in the Δ*tpiA* mutant at exponential phase.** Bacteria were cultured in LB at 37 °C till $OD_{600}$ reached 0.5. (A) mRNA levels of the QS signaling molecule synthesis genes, including *lasI*, *rhlI* and *pqsA* were determined by RT-qPCR. (B) Promoter activities of *lasI*, *rhlI* and *pqsA* were detected by β-galactosidase activity assay. (C) 3-Oxo-$C_{12}$-HSL, $C_4$-HSL and the PQS signal molecules (PQS and HHQ) levels in the supernatants of indicated strains were measured with corresponding reporter strains using β-galactosidase activity assay. Data represent the mean ± standard deviation of the results from three samples. \*\*\*\*, $P < 0.0001$; \*\*\*, $P < 0.001$; \*\*, $P < 0.01$; \*, $P < 0.05$ by Student's *t* test.
(TIF)

**S3 Fig. TpiA controls the Las system through RsaL.** Bacteria were cultured in LB at 37 °C till OD$_{600}$ reached 2.5. mRNA levels of *lasI* and *lasR* were determined by RT-qPCR. Data represent the mean ± standard deviation of the results from three samples. ***, $P < 0.001$; **, $P < 0.01$; *, $P < 0.05$ by Student's *t* test.
(TIF)

**S4 Fig. Transcription of *lasR* in the Δ*tpiA* mutant.** Bacteria were cultured in LB at 37 °C till OD$_{600}$ reached 2.5. (A) mRNA levels of *lasR* were determined by RT-qPCR. (B) Promoter activities of *lasR* were determined by β-galactosidase activity assay. Data represent the mean ± standard deviation of the results from three samples. ****, $P < 0.0001$; ***, $P < 0.001$ by Student's *t* test.
(TIF)

**S5 Fig. Acetylation of CspC.** Bacteria containing *cspC-gst* or *gst* driven by P$_{cspC}$ were cultured in LB at 37 °C till OD$_{600}$ reached 2.5. Acetylation and total amounts of the purified CspC-GST (A) and GST (B) were determined by Western Blot.
(TIF)

**S6 Fig. Identification of the regulator of *cspC*.** (A) Identification of candidate proteins binding to the P$_{cspC}$ promoter. The biotin-labeled P$_{cspC}$ promoter fragment was incubated with the cell lysate of PA14 or the Δ*tpiA* mutant, followed by purification with streptavidin-conjugated beads. The protein bands different from the control samples (cell lysates without DNA fragment) are indicated with arrows. (B) mRNA levels of the candidate regulatory genes were determined by RT-qPCR. (C) mRNA levels of *cspC* in transposon insertion mutants of the candidate genes were determined by RT-qPCR. (D) mRNA levels of *cspC* in indicated strains were determined by RT-qPCR. (E) Promoter activities of *cspC* were detected by β-galactosidase activity assay. Data represent the mean ± standard deviation of the results from three samples. ***, $P < 0.001$; **, $P < 0.01$; *, $P < 0.05$ by Student's *t* test.
(TIF)

**S7 Fig. Purification of AgtR from *E. coli*.** AgtR fused with a N-terminus 6×His tag and a solubilizing tag SUMO was purified with Ni-affinity chromatography and subjected to SDS-PAGE analysis and Coomassie blue staining. The purified protein is indicated by an arrowhead. M, protein marker.
(TIF)

**S8 Fig. Identification of the transcription start site of *cspC*.** The transcriptional start site was determined by 5'-RACE assay and indicated by a red arrow.
(TIF)

**S9 Fig. Competition between *S. aureus* and *P. aeruginosa* strains.** Bacteria were grown overnight in LB at 37 °C. Indicated *P. aeruginosa* strains and *S. aureus* RN4220 were mixed at a 1:30 ratio and cocultured in LB at 37 °C for 14 h. The competitive indexes represent the ratio of *P. aeruginosa* to *S. aureus*. Data represent the mean ± standard deviation of the results from three samples. **, $P < 0.01$; *, $P < 0.05$ by Student's *t* test.
(TIF)

**S10 Fig. The AgtR-CspC pathway is involved in sensing and competition against other bacteria.** Bacteria were grown overnight in LB at 37 °C. (A) *P. aeruginosa* and *S. agalactiae* were mixed at a 1:3 ratio and cocultured in LB at 37 °C for 6 h. The competitive indexes represent the ratio of *P. aeruginosa* to *S. agalactiae*. (B) *P. aeruginosa* and *S. epidermidis* were mixed at a 1:30 ratio and cocultured in LB at 37 °C for 6 h. The competitive indexes represent the

ratio of *P. aeruginosa* to *S. epidermidis*. Data represent the mean ± standard deviation of the results from three samples. \*\*\*, $P < 0.001$; \*\*, $P < 0.01$; \*, $P < 0.05$ by Student's *t* test.
(TIF)

**S1 Table. Culture time and concentrations of the bacteria at the corresponding OD$_{600}$.**
(DOCX)

**S2 Table. Primers used in this study.**
(DOCX)

**S3 Table. Plasmids and strains used in this study.**
(DOCX)

**S4 Table. Mass spectrometric result of the protein bands in the DNA pull-down assay.**
(DOCX)

**S1 Text. Supplemental Methods. Strains and plasmid construction.**
(DOCX)

**S1 Data. Minimal data set.**
(XLSX)

## Author contributions

**Conceptualization:** Junze Qu, Weihui Wu.

**Data curation:** Junze Qu, Liwen Yin, Shanhua Qin, Xiaomeng Sun, Xuetao Gong, Shouyi Li, Xiaolei Pan.

**Formal analysis:** Junze Qu, Yongxin Jin, Zhihui Cheng, Shouguang Jin, Weihui Wu.

**Funding acquisition:** Shouyi Li, Yongxin Jin, Zhihui Cheng, Weihui Wu.

**Investigation:** Junze Qu, Liwen Yin, Shanhua Qin, Xiaomeng Sun, Xuetao Gong, Shouyi Li, Xiaolei Pan.

**Methodology:** Junze Qu, Yongxin Jin, Zhihui Cheng, Shouguang Jin, Weihui Wu.

**Project administration:** Weihui Wu.

**Resources:** Yongxin Jin, Zhihui Cheng, Shouguang Jin, Weihui Wu.

**Software:** Junze Qu, Liwen Yin, Shanhua Qin.

**Supervision:** Weihui Wu.

**Validation:** Junze Qu, Weihui Wu.

**Visualization:** Junze Qu, Weihui Wu.

**Writing – original draft:** Junze Qu, Weihui Wu.

**Writing – review & editing:** Junze Qu, Shouguang Jin, Weihui Wu.

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
