## [Decision Letter · Decision Letter 0]

3 Dec 2024

PPATHOGENS-D-24-02248Identification of the Pseudomonas aeruginosa AgtR-CspC-RsaL pathway that controls Las quorum sensing in response to metabolic perturbation and Staphylococcus aureusPLOS PathogensDear Dr. Wu,

Thank you for submitting your manuscript to PLOS Pathogens. After careful consideration, we feel that it has merit but does not fully meet PLOS Pathogens's publication criteria as it currently stands. Therefore, we invite you to submit a revised version of the manuscript that addresses the points raised during the review process. The reviewers raised significant concerns about over interpretation of the results, especially related to the regulation of *lasR * in a *tipA * mutant and the role of TipA and CspC in the direct regulation of *rsaL* . As evidenced by the the comments of reviewers 2 and 3, there is insufficient data to make certain claims in the manuscript and more experiments with additional controls are required. Additionally, the manuscript must be heavily edited for clarity and to provide better context for the experimental procedures and the results.

Please submit your revised manuscript within 60 days Feb 01 2025 11:59PM. If you will need more time than this to complete your revisions, please reply to this message or contact the journal office at plospathogens@plos.org. Please include the following items when submitting your revised manuscript: * A rebuttal letter that responds to each point raised by the editor and reviewer(s). You should upload this letter as a separate file labeled 'Response to Reviewers '. This file does not need to include responses to any formatting updates and technical items listed in the 'Journal Requirements' section below. * A marked-up copy of your manuscript that highlights changes made to the original version. You should upload this as a separate file labeled 'Revised Manuscript with Track Changes '. * An unmarked version of your revised paper without tracked changes. You should upload this as a separate file labeled 'Manuscript '. If you would like to make changes to your financial disclosure, competing interests statement, or data availability statement, please make these updates within the submission form at the time of resubmission. Guidelines for resubmitting your figure files are available below the reviewer comments at the end of this letter. We look forward to receiving your revised manuscript. Kind regards, Jon PaczkowskiAcademic EditorPLOS PathogensMatthew WolfgangSection EditorPLOS PathogensMichael Malim

Editor-in-Chief

PLOS Pathogens

orcid.org/0000-0002-7699-2064

**Journal Requirements:**

At this stage, the following Authors/Authors require contributions: Junze Qu, Linwen Yin, Shanhua Qin, Xiaomeng Sun, Xuetao Gong, Shouyi Li, Xiaolei Pan, Yongxin Jin, Zhihui Cheng, Shouguang Jin, and Weihui Wu. Please ensure that the full contributions of each author are acknowledged in the "Add/Edit/Remove Authors" section of our submission form.

- TM on page: Lines: 427..

3) Please amend your detailed Financial Disclosure statement. This is published with the article. It must therefore be completed in full sentences and contain the exact wording you wish to be published. Please ensure that the funders and grant numbers match between the Financial Disclosure field and the Funding Information tab in your submission form. Note that the funders must be provided in the same order in both places as well.

**Reviewers' Comments:** Reviewer's Responses to Questions

**Part I - Summary**

Reviewer #1: In this study, the authors demonstrated that the mutation of tpiA results in increased expression of QS genes, such as lasI, rhlI, and pqsA, as well as the production of QS signals. This upregulation is due to CspC mediated translational repression of rsaL in the tpiA mutant. The study also identifies AgtR as a regulator that upregulates cspC in the ΔtpiA mutant, and demonstrates that AgtR directly binds to the cspC promoter region to activate transcription. The authors also linked the AgtR-CspC pathway to the bacterial response to GlcNAc and competition with Staphylococcus aureus, providing a broader ecological context to their findings. The study concludes that the AgtR-CspC pathway plays a significant role in sensing environmental cues and in bacterial competition. Overall, the study is well-executed and provides valuable insights into the regulatory mechanisms of QS systems in P. aeruginosa.

Reviewer #2: In this research manuscript, Qu et al. present new data on the AgtR-CspC-RsaL-LasI pathway, and establish links between metabolism, quorum sensing (QS) and interspecies competition in P. aeruginosa PA14.

Appropriate experiments were conducted to investigate the subject and they appear well executed. Unfortunately, the main weakness of this paper is the writing, making for a frustrating read. It lacks important details in the introduction and the methodology, and a lot of references are inaccurate or misplaced. Also, the context of each result section and experiment is poorly introduced.

Also why look at all three QS system and not mention the implication of the rhl and PQS systems in the model? The authors focus on pyocyanin because it is overproduced in the tpiA mutant. But they also show induced expression of genes related to the las, rhl and PQS systems which together regulate a lot of factors which could also impact interaction with Staphylococcus aureus.

RsaL, which is mentioned in the paper to be regulated by CspC, is also a direct known regulator of the phz genes and thus pyocyanin production. Nothing is mentioned about this even if this would simply explain the enhanced pyocyanin production.

The pathway involved in sensing and competition against S. aureus: considering GlcNAc is a component of peptidoglycan, this response could be non-specific and a mechanism of defense against a wide variety of bacterium. It would be interesting to test the upregulation of this pathway against a variety of different bacterium and not only S. aureus.

The manuscript needs to be substantially rewritten for clarity and better context and with the right references throughout.

Reviewer #3: This manuscript connects the regulators AgtR and CspC with phenotypes associated with deletions in the gene encoding TpiA, a central metabolism enzyme that converts between dihydroxyacetone phosphate and glycerol-3-phosphate (connecting glycolysis with glycerol metabolism). AgtR is a peptidoglycan-responsive regulator that has regulatory effects on several virulence factors, including quorum sensing. CspC is a recently described translation regulator that seems to have a number of different targets in the cell, several of which have roles in virulence. The studies show that some regulatory changes that occur in a ∆tpiA mutated strain occur through the AgtR-CspC pathway.

The results add to the current understanding of AgtR and CpsC regulatory functions and how these regulators connect with other networks in the cell. The results could be helpful for understanding changes in central metabolism or virulence regulation.

However, there were some major clarity and data interpretation issues in the manuscript that need addressed, as described below.

**Part II – Major Issues: Key Experiments Required for Acceptance**

Reviewer #1: 1. The phz1 and phz2 operons share a high degree of homology. Which phzA mRNA level did the authors examine? I noticed that in the supplementary information, qRT-PCR primers for phzB1 but not phzA were listed. Which gene was tested? The authors need to clarify whether the primers are operon-specific, or the overall phzA or phzB levels (e.g. phzA1 + phzA2) were measured.

2. It seems AgtS-AgtR senses multiple ligands. Are there structural similarities between these ligands?

Reviewer #2: When working on QS, it is important to consider timing of experimentations. The results obtained clearly show higher expression of QS-related genes. However, only one time point was chosen, precluding assessment of the QS effect. Does the tipA mutant grows similarly as the wildtype? Are results the same at other phases of growth? It would have been very useful to see this. The authors should at least demonstrate that the growth curve of ΔtipA is the same as that of the WT.

L139 : The title of the section named ‘’TpiA controls the QS systems through RsaL’’ is misleading. In this section, the authors indeed demonstrate that TpiA influences the transcription and translation of rsaL, a target of LasR, the key transcriptional regulator of the Las system. To clarify the direct effect of TpiA on the Las system, it would have been valuable to include data on the regulation of LasR itself. For example, assessing a reporter construct including lasR and its native promotor could verify whether TpiA directly impacts LasR, given the numerous regulators involved in controlling LasR expression. Perhaps a double mutant of tpiA and rsaL would have provided insight to look at the combined effect of these two genes on LasR regulation. Also, the impact on the other two QS systems is not described here.

In the introduction, pyocyanin is mentioned as being important in the Pa and Sa interaction and the authors show that it is overexpressed in co-cultures in presence of Sa or with GlcNAc. This induction is lost in both the agtR and cspC mutants. This does not prove that pyocyanin is responsible for the competition observed. It could be another QS regulated factor. Comparing with a double agtR phz mutant, unable to produce pyocyanin would answer that question.

Reviewer #3: The interpretation of results with the rsaL transcription reporter is unclear. rsaL transcription is activated by LasR; if the LasR-I system is induced then rsaL will also be impacted. It seems likely the ∆tpiA mutation impacts rsaL transcription by modulating QS but it is not clear if this was the proposed mechanism.

It seems that the authors are proposing a mechanistic model whereby the ∆tpiA mutation leads to translational repression of RsaL, which leads to higher levels of QS. However, this model was never directly tested (e.g. by using an rsaL deletion or by examining QS activation in bacteria where RsaL regulation was unlinked from the ∆tpiA mutation).

Fig. 6C – it is unclear that the effects of the ∆tpiA and ∆cspC mutations are through the changes in RsaL levels. To test this directly, an rsaL deletion mutant would need to be tested, or by unlinking RsaL regulation from the ∆tpiA mutation).

Fig. 10: what is RN4220? There is no mention of this in the legend or methods, unless something was missed.

**Part III – Minor Issues: Editorial and Data Presentation Modifications**

Reviewer #1: 1. “illustated” should be “illustrated” in the discussion section.

2. The language used in the manuscript is generally clear, but some sentences are quite dense and could be broken down for better readability, e.g. line 256-258.

3. Fig 1, 2, 3, 5, the authors should clarify which samples were compared.

4. Line 269, what does “coenzyme” mean here?

5. Line 270, it is better to replace “agtA” with “the agtABCD operon”.

Reviewer #2: Correct the following molecules throughout the text so that the number of atoms are in subscript : 3-oxo-C12-HSL; C4-HSL

Line 48 : Demonstrate

Line 53 : co-exists

Line 70, A better reference for this should be Mavrodi et al. J. Bacteriol. 2001; 183(21) : 6454-65

Line 71, the wrong reference is cited. A suggestion would be Latifi et al. Mol Microbiol. 1995;17(2)333-43

Ln 78-81: This sentence is not clear. PqsH is implicated only in the synthesis of PQS and it seems the sentence was written as if it were HHQ. Also, the chosen references are not appropriate for describing this system.

Line 81: Again, these references are not appropriate.

Line 84 Suppresses is a strong word. Maybe inhibit?

Line 85: Again, wrong reference.

Line 126 and others: it should be RT-qPCR instead of qRT-PCR.

Line 142: Again, this is not the best reference for this claim.

Figure 2C: Description is lacking in the legend. Miller Units does not say anything on what was used to measure the signals. We should not have to go to the material and method to understand what we see. Moreover, this section is not well described in material and method. Firstly, once again, the reference cited has nothing to do with what is described and we need to refer to Supplementary Tables to try and understand which tool was used.

Figure 2C: it is likely your reporter detects HHQ as well as PQS and what you measure is a combination of both. Know the limits of the methods used.

Line 156: where are the references for those previous studies?

You first mention the role of AgtR at line 230. It would have been important to include a more through description of this regulator and its context in the introduction, as it is only briefly mentioned at line 93. Also, maybe include more information as to why you chose AgtR to pursue your experiments, and not GntR or PA0756, who are also upregulated in a tpiA mutant.

Fig 7B: it is hard to distinguish between the different levels of greys for each strain.

Lline 306: what statistical method was used for qPCR ? Was it 2ΔΔCT ? Is rspL known to be a good internal control ? Is there a reference ? If not, authors should show the levels of expression in the different backgrounds to demonstrate it is similar.

Line 333: should still be a better description of the tool used to quantity PQS than Pseudomonas reporter strain.

Line 454 : ratio and (add space between the two words)

Fig 10: If you use RN4220 in the figure, it should be described in the legend.

Table S2: details should be added on strains listed as Laboratory stocks. Were theses previously published ? If not the details on their construction should be included. Were they received from another lab ? The reference should be added.

Competition assay: it is not clear how were chosen the ratio for competition and what is the impact of this ratio on the results.

Reviewer #3: Lines 71-74: It is not clear what is meant by QS enabling responses to other environmental signals or species composition. Consider rewording the description of QS to “a bacterial cell-cell communication system that enables cells to respond to changes in cell density” as this is less confusing.

Figures 1 and 2 can be combined, or Fig. 1 could be moved to the supplement.

Lines 194-195: it is unclear what is meant by, “among which the mRNA levels of gntR, PA0756 and agtR were elevated in the ΔtpiA mutant (S2B Fig).” It seems from the figure that the mRNA levels were measured for each of these genes in the ∆tpiA mutant but the way this sentence is written it sounds like the mRNA was somehow measured during the biotin pulldown experiment.

Line 224: the text needs to modified to clarify that lacZ is measuring cpsC expression levels

PLOS authors have the option to publish the peer review history of their article (what does this mean? ). If published, this will include your full peer review and any attached files.

**Do you want your identity to be public for this peer review?** For information about this choice, including consent withdrawal, please see our Privacy Policy .

Reviewer #1: No

Reviewer #2: No

Reviewer #3: No

---

## [Decision Letter · Decision Letter 1]

11 Mar 2025

PPATHOGENS-D-24-02248R1

Identification of the Pseudomonas aeruginosa AgtR-CspC-RsaL pathway that controls Las quorum sensing in response to metabolic perturbation and Staphylococcus aureus

PLOS Pathogens

Dear Dr. Wu,

Thank you for submitting your manuscript to PLOS Pathogens. After careful consideration, we feel that it has merit but does not fully meet PLOS Pathogens's publication criteria as it currently stands. Therefore, we invite you to submit a revised version of the manuscript that addresses the points raised during the review process.

It is important that you address the one remaining comment from Reviewer 2 in the main body of the text.

Please submit your revised manuscript within 30 days May 10 2025 11:59PM. If you will need more time than this to complete your revisions, please reply to this message or contact the journal office at plospathogens@plos.org. Please include the following items when submitting your revised manuscript:

We look forward to receiving your revised manuscript.

Kind regards,

Jon Paczkowski

Academic Editor

PLOS Pathogens

Matthew Wolfgang

Section Editor

PLOS Pathogens

Sumita Bhaduri-McIntosh

Editor-in-Chief

PLOS Pathogens

orcid.org/0000-0003-2946-9497

Michael Malim

Editor-in-Chief

PLOS Pathogens

orcid.org/0000-0002-7699-2064

**Journal Requirements:**

At this stage, the following Authors/Authors require contributions: Junze Qu, Liwen Yin, Shanhua Qin, Xiaomeng Sun, Xuetao Gong, Shouyi Li, Xiaolei Pan, Yongxin Jin, Zhihui Cheng, Shouguang Jin, and Weihui Wu. Please ensure that the full contributions of each author are acknowledged in the "Add/Edit/Remove Authors" section of our submission form.

2) The following file is currently uploaded as file type 'Other', which is not viewable by the reviewers: "minimal data set.xlsx. " Please change the file type to 'Supporting Information' and include a legend in the manuscript.

**Reviewers' Comments:**

Reviewer's Responses to Questions

**Part I - Summary**

Reviewer #1: The authors have addressed the previous comments from the reviewer.

Reviewer #2: I am mostly satisfied with the revised version.

However, I still have one concern that must be addressed. About my question on any growth rate difference between the tpiA mutant and the WT, which might influence the quorum sensing response, the authors confirm that the mutant is growing slower and refer me to the supplementary data of a previously published paper. I am not fully satisfied by this answer. I checked, and the growth defect is actually important. It must be clearly mentioned in the present manuscript that the mutant has a growth defect. The authors state that they matched cultures by OD600 instead of timing, this likely represents several hours of difference in timing of sampling. While I agree that this could be OK when looking at gene expression, the same does not goes for measurement of accumulated metabolites, for instance. The authors should simply mention that because of the growth defect, cultures we sampled at corresponding OD600 (the growth curves in the previous paper does allow to verify if this corresponds to the same growth stage, so the authors cannot claim that, as we do not know if the mutant can eventually reach the same maximal cell density… In fact, according to the growth curves shown in the previous paper, it is very likely that it never reaches the same OD600 as the WT). The authors should keep in mind that OD600 measurements are only a proxy, and the actual information must be obtained by counting CFUs, anyway.

Reviewer #3: Prior criticisms were sufficiently addressed

**Part II – Major Issues: Key Experiments Required for Acceptance**

Reviewer #1: (No Response)

Reviewer #2: (No Response)

Reviewer #3: (No Response)

**Part III – Minor Issues: Editorial and Data Presentation Modifications**

Reviewer #1: (No Response)

Reviewer #2: (No Response)

Reviewer #3: (No Response)

PLOS authors have the option to publish the peer review history of their article (what does this mean? ). If published, this will include your full peer review and any attached files.

**Do you want your identity to be public for this peer review?** For information about this choice, including consent withdrawal, please see our Privacy Policy .

Reviewer #1: No

Reviewer #2: No

Reviewer #3: No

**Figure resubmission:**
---

## [Editor Report · Decision Letter 2]

19 Mar 2025

Dear Mr. Wu,

We are pleased to inform you that your manuscript 'Identification of the Pseudomonas aeruginosa AgtR-CspC-RsaL pathway that controls Las quorum sensing in response to metabolic perturbation and Staphylococcus aureus' has been provisionally accepted for publication in PLOS Pathogens.

Best regards,

Jon Paczkowski

Academic Editor

PLOS Pathogens

Matthew Wolfgang

Section Editor

PLOS Pathogens

Sumita Bhaduri-McIntosh

Editor-in-Chief

PLOS Pathogens

orcid.org/0000-0003-2946-9497

Michael Malim

Editor-in-Chief

PLOS Pathogens

orcid.org/0000-0002-7699-2064
---

## [Editor Report · Acceptance letter]

Dear Mr. Wu,

We are delighted to inform you that your manuscript, "Identification of the Pseudomonas aeruginosa AgtR-CspC-RsaL pathway that controls Las quorum sensing in response to metabolic perturbation and Staphylococcus aureus," has been formally accepted for publication in PLOS Pathogens.

Best regards,

Sumita Bhaduri-McIntosh

Editor-in-Chief

PLOS Pathogens

orcid.org/0000-0003-2946-9497

Michael Malim

Editor-in-Chief

PLOS Pathogens

orcid.org/0000-0002-7699-2064